# Enhancing Transfer Learning with Flexible Nonparametric Posterior Sampling

**Hyungi Lee**[*1]    **Giung Nam**[*1]    **Edwin Fong**[2]    **Juho Lee**[1,3]
[1]KAIST AI    [2]The University of Hong Kong    [3]AITRICS
{lhk2708, giung, juholee}@kaist.ac.kr, chefong@hku.hk

## Abstract

Transfer learning has recently shown significant performance across various tasks involving deep neural networks. In these transfer learning scenarios, the prior distribution for downstream data becomes crucial in Bayesian model averaging (BMA). While previous works proposed the prior over the neural network parameters centered around the pre-trained solution, such strategies have limitations when dealing with distribution shifts between upstream and downstream data. This paper introduces nonparametric transfer learning (NPTL), a flexible posterior sampling method to address the distribution shift issue within the context of nonparametric learning. The nonparametric learning (NPL) method is a recent approach that employs a nonparametric prior for posterior sampling, efficiently accounting for model misspecification scenarios, which is suitable for transfer learning scenarios that may involve the distribution shift between upstream and downstream tasks. Through extensive empirical validations, we demonstrate that our approach surpasses other baselines in BMA performance.

## 1 Introduction

In Bayesian deep learning, we regard the parameters of a deep neural network as random variables. Instead of optimizing for a single-point estimate of these parameters, this approach involves inferring the posterior distribution of these parameters given the provided training data and predefined parameter prior distribution. After we have the posterior distribution, we make predictions through Bayesian model averaging (BMA). BMA entails computing predictions from multiple parameter values and weighting them based on their respective densities within the posterior. BMA effectively integrates both data uncertainty and model uncertainty into the prediction process, leading to more accurate and resilient predictions (Hoeting et al., 1999).

The success of Bayesian deep learning often depends on the choice of the prior distribution. While it is common practice to employ a simple zero-mean Gaussian prior for neural network parameters, there is an ongoing discussion regarding the adequacy of these zero-mean Gaussian priors (Wenzel et al., 2020; Fortuin et al., 2022). Meanwhile, in the context of transfer learning scenarios, these concerns about prior configurations are further intensified. The fundamental idea behind the transfer learning process is that when model parameters are pre-trained using sufficiently extensive and versatile upstream data, they inherently capture biases related to the data modality, which can be beneficial in related downstream tasks (Mikolov et al., 2013; Girshick et al., 2014). In this case, there is a doubt that the Gaussian prior over neural network parameters can sufficiently capture solely the *"prior knowledge"* embedded in the upstream data.

To address this issue, we directed our attention to a nonparametric posterior sampling method called Bayesian nonparametric learning (NPL; Lyddon et al., 2018; Fong et al., 2019). NPL enables the use of statistical models without assuming the model is true. It utilizes a nonparametric prior, such as a mixture of Dirichlet Processes (Antoniak, 1974), centered around a parametric model then updates a nonparametric posterior for the parameters of the parametric model. The NPL approach effectively accounts for the model misspecification by an implicitly defined prior. This prior is chosen without

---

* Equal contribution

any reliance on the specific model of interest, which is suitable in the context of a misspecified model as we do not have confidence in the existence of a true model.

Our primary contribution is utilizing the NPL approach, which effectively accounts for the model misspecification, within the context of transfer learning. While the fundamental premise of transfer learning is that the *"prior knowledge"* obtained from upstream data is advantageous for downstream tasks, empirical studies have indicated that prior regularization centered around pre-trained model in the weight space can hinder the downstream learning process in cases where there are mismatches between the upstream knowledge and downstream tasks (Xuhong et al., 2018; Wan et al., 2019; Li et al., 2020). In this regard, we were intrigued by the prospective benefits presented by NPL's efficacy in addressing model misspecification, especially in transfer learning situations where there can be a distribution shift between upstream and downstream data. This is done by combining the upstream and downstream information in a *nonparametric* fashion, resembling a form of weighted summation between the downstream dataset and the pseudo dataset.

We summarize our contribution as follows:

- We proposed nonparametric transfer learning (NPTL): a posterior sampling method that adapts the NPL method into the transfer learning scenario where both the parameter of interest and the downstream dataset are very large.

- Our proposed posterior sampling algorithm can be easily parallelized. We highlight that such parallelization is not straightforward for stochastic gradient Markov Chain Monte Carlo (MCMC) methods, which are commonly used in Bayesian deep learning for posterior sampling, due to their sequential nature (Neiswanger et al., 2014; De Souza et al., 2022).

- We empirically validate that the proposed algorithm shows better performance compared to the other baseline posterior sampling methods.

## 2 PRELIMINARIES

### 2.1 BAYESIAN DEEP LEARNING

In Bayesian inference, the goal is to sample from the posterior distribution $p(\boldsymbol{\theta}|\mathcal{D})$ over the parameters $\boldsymbol{\theta}$ after observing the data $\mathcal{D}$. The prediction for a new datum $x$ is then given by Bayesian model averaging (BMA),

$$p(y|x, \mathcal{D}) = \int p(y|x, \boldsymbol{\theta})p(\boldsymbol{\theta}|\mathcal{D})\mathrm{d}\boldsymbol{\theta}, \tag{1}$$

which we can approximate by Monte Carlo integration $p(y|x, \mathcal{D}) \approx \sum_{m=1}^{M} p(y|x, \boldsymbol{\theta}_m)/M$ with posterior samples $\boldsymbol{\theta}_1, ..., \boldsymbol{\theta}_M \sim p(\boldsymbol{\theta}|\mathcal{D})$. In practice, we typically introduce several sampling methods for $p(\boldsymbol{\theta}|\mathcal{D})$ to compute the BMA integration since it cannot be expressed in closed form for modern deep neural networks.

In order to work with the posterior $p(\boldsymbol{\theta}|\mathcal{D})$, it is essential to first establish the prior $p(\boldsymbol{\theta})$. The most common choice of the uninformed prior distribution is a zero mean isotropic Gaussian, i.e. $p(\boldsymbol{\theta}) = \mathcal{N}(\boldsymbol{\theta}; \mathbf{0}, \sigma^2\mathbf{I})$, which is equivalent to $L_2$ weight decay regularization (Krogh & Hertz, 1991). A straightforward way to enhance such a prior is to select the mean of the isotropic Gaussian with a pre-determined value (Chelba & Acero, 2006; Daumé III, 2007; Grachten & Chacón, 2017; Xuhong et al., 2018), i.e. $p(\boldsymbol{\theta}) = \mathcal{N}(\boldsymbol{\theta}; \boldsymbol{\theta}_{\text{prior}}, \sigma^2\mathbf{I})$, where $\boldsymbol{\theta}_{\text{prior}}$ is some informative weight with condensed information of the source task (e.g. MAP solution pre-trained on the source task). Recently, Shwartz-Ziv et al. (2022) proposed a method to build an informative prior from a pre-trained model, where they compute *both* mean and covariance matrix of the Gaussian prior using SWA-Gaussian (Maddox et al., 2019) procedure. However, none of these methods take into account the downstream dataset when creating the weight prior distribution.

### 2.2 TRANSFER LEARNING

Modern deep learning algorithms often employ the *pre-training and fine-tuning paradigm* (He et al., 2019): pre-train a model from scratch on a large source dataset, then fine-tune the model for a specific target task. This is based on the belief that there is some *transferable information* between

source and target tasks, i.e. we expect features learned from source data to be useful on target tasks. Compared to training from scratch on a target task where the optimization starts from a randomly initialized parameter, transfer learning starting from the pre-trained parameter converges quicker to a reasonable solution. While it is common to fine-tune all parameters in a pre-trained model, depending on the target task and computational resources available, it may be possible to fine-tune only a subset of parameters while keeping the rest fixed. An example of this is *linear probing*, where the feature extractor is frozen, and only the last linear layer is newly trained for the target task.

## 2.3 Nonparametric learning

We assume that a downstream dataset $\mathcal{D} := (x_i, y_i)_{i=1}^n$ is drawn i.i.d. from some data distribution $F_0$, and we are interested in a class of parametric models $\mathcal{F}_\Theta := \{F_\theta \mid \theta \in \Theta\}$ where $F_\theta$ denotes the model data with $\theta \in \Theta \subseteq \mathbb{R}^p$ (e.g. a neural network with parameter $\theta$). Here, we do not assume that our target distribution $F_0$ belongs to $\mathcal{F}_\Theta$, indicating the possibility of model misspecification.

The NPL framework defines a parameter of interest as a functional of $F_0$, that is, $\theta_0 = \theta(F_0)$ where

$$\theta(F_0) = \arg\min_\theta \int \ell(\theta; x, y) \, dF_0(x, y), \tag{2}$$

and $\ell(\theta; x, y)$ is a loss function of interest. In supervised learning, we set $\ell(\theta; x, y) = -\log f_\theta(y \mid x, \theta)$ as the negative log density of $F_\theta$, which gives $\theta_0 = \arg\min_\theta D_{\mathrm{KL}}[F_0 \| F_\theta]$ (Walker, 2013). Since we do not know $F_0$, we treat it as a random measure $F$, and elicit a Dirichlet process (DP; Ferguson, 1973) prior following Fong et al. (2019), that is $F \sim \mathrm{DP}(\alpha, F_\pi)$. Here, $F_\pi$ is a centering probability measure acting as our prior guess on $F_0$, and $\alpha \in \mathbb{R}_+$ is a scalar hyperparameter indicating the strength of this prior belief. By conjugacy, the posterior of $F$ given $\mathcal{D}$ is also a DP:

$$F \mid \mathcal{D} \sim \mathrm{DP}\left(\alpha + n, \frac{\alpha}{\alpha + n} F_\pi + \frac{1}{\alpha + n} \sum_{i=1}^n \delta_{(x_i, y_i)}\right).$$

The NPL posterior $\tilde{p}(\theta \mid \mathcal{D})$ is then the pushforward measure of $p(F \mid \mathcal{D})$ through (2), from which one can sample by drawing $F^{(j)} \sim p(F \mid \mathcal{D})$ and computing $\theta^{(j)} = \theta\left(F^{(j)}\right)$. Draws from $p(F \mid \mathcal{D})$ can be carried out using a skeleton of $T$ atoms from $F_\pi$ and appropriate Dirichlet weights, which we outline later. When $\ell$ is highly non-convex, Fong et al. (2019) explores two possible heuristics for feasible sampling. If one is interested in sampling from multiple modes, the global minima in (2) can be estimated via random restart optimization with multiple random initializations. Alternatively, one may target a local mode by initializing each optimization at some fixed $\theta_{\mathrm{init}}$ of interest around which we want to quantify uncertainty. Finally, Lyddon et al. (2018); Fong et al. (2019) show that the NPL posterior is robust to model misspecification through the adoption of a nonparametric model, and gives asymptotically superior predictions to the regular Bayesian posterior on $\theta$.

## 3 Nonparametric Transfer Learning

In this section, we provide details on our proposed approach for posterior sampling called NPTL, tailored for common transfer learning scenarios involving a single pre-trained parameter. The key feature of these situations is that downstream solutions are typically found near the pre-trained parameter (Neyshabur et al., 2020). Consequently, this aligns with the second scenario described in § 2.3, which focuses on exploring a local mode centered around the pre-trained parameter.

## 3.1 Base measure

To form our initial estimate $F_\pi$ in the DP prior, it is essential to incorporate our prior understanding of the true data distribution $F_0$. This prior knowledge can be derived from the upstream task in transfer learning scenarios, which we outline below. Formally, we define an informative base measure $F_\pi(x, y)$ by utilizing the empirical distribution of the upstream dataset $\{x_i^{\mathrm{up}}, y_i^{\mathrm{up}}\}_{i=1}^{n_{\mathrm{up}}}$,

$$F_\pi(x, y) = \frac{1}{n_{\mathrm{up}}} \sum_{i=1}^{n_{\mathrm{up}}} \delta_{(x_i^{\mathrm{up}}, y_i^{\mathrm{up}})}(x, y),$$

---

**Algorithm 1** Nonparametric Transfer Learning.

---

**Require:** Pre-trained feature extractor parameters $\phi^*$ and the linear-probed head $\mathbf{W}^*$ for downstream data $\{x_i, y_i\}_{i=1}^n$. The linear-probed model $f_{\text{probed}}$ is parameterized by $(\phi^*, \mathbf{W}^*)$.
**Ensure:** Downstream posterior samples $\{\boldsymbol{\theta}^{(m)}\}_{m=1}^M$.

1: Define $f_\pi(x) := f_{\text{probed}}(x)$.
2: **for** $m = 1$ **to** $M$ **do**
3:     Generate prior pseudo-samples $(x_i, f_\pi(x_i))_{i=1}^n$.
4:     Randomly construct index mapping function $u : [n] \to [L] \times [L]$
5:     Draw $(v_{1:L}^{(m)}, \tilde{v}_{1:L}^{(m)}) \sim 2L * \text{Dir}(1, \dots, 1, \alpha/n, \dots, \alpha/n)$.
6:     Replace $(w_1, \dots, w_n, \widetilde{w}_1, \dots, \widetilde{w}_n) = (v_{u(1)[1]}, \dots, v_{u(n)[1]}, \tilde{v}_{u(1)[2]}, \dots, \tilde{v}_{u(n)[2]})$
7:     Optimize $\boldsymbol{\theta}^{(m)} = \arg\min_{\boldsymbol{\theta}} \left\{ \sum_{j=1}^n w_j^{(m)} \ell(\boldsymbol{\theta}; x_j, y_j) + \sum_{k=1}^n \widetilde{w}_k^{(m)} \ell(\boldsymbol{\theta}; x_k, f_\pi(x_k)) \right\}$.
8: **end for**

---

where $\delta_{(x,y)}$ denotes a discrete point measure at $(x, y)$. However, in many real-world machine learning scenarios, this approach faces challenges for two primary reasons. Firstly, the upstream datasets are often kept private and undisclosed, e.g. JFT-300M and JFT-3B datasets (Sun et al., 2017). Secondly, handling such extensive datasets can demand substantial memory resources, making it unfeasible to directly build the empirical distribution.

To overcome these issues, we introduce an alternative approach that solely depends on pre-trained parameters and downstream data for constructing an informative base measure. Specifically, we first partition the model parameters $\boldsymbol{\theta}$ into feature extractor parameters $\phi$ and task-specific head parameters $\mathbf{W}$, i.e. $\boldsymbol{\theta} = (\phi, \mathbf{W})$, to address the variability in output dimensions across different tasks. Next, we utilize linear probing to transfer the upstream knowledge to the downstream setting, as the pre-trained feature extractor parameters $\phi^*$ should capture the essence of the upstream data generation process. Using the linear-probed model $f_{\text{probed}}$ and downstream inputs $\{x_i\}_{i=1}^n$, we define the informative base measure $F_\pi(x, y)$ as

$$F_\pi(x, y) = \frac{1}{n} \sum_{i=1}^n \delta_{(x_i, f_{\text{probed}}(x_i))}(x, y). \tag{3}$$

Our DP prior on $F_0$ then becomes $\text{DP}(\alpha, F_\pi)$, where $\alpha$ is the strength of our belief. The value of $\alpha$ is chosen through a validation set to regulate the proper degree of our belief, outlined below.

**Empirical Bayes.** We highlight that both $F_\pi$ and $\alpha$ in the DP prior are dependent on the downstream data $\mathcal{D}$, which indicates that our method can be understood as an *empirical Bayes* approach (Robbins, 1956). In particular, we address two key aspects: 1) the choice of a loss function and the data for training $\mathbf{W}$ and 2) the selection of an appropriate $\alpha$. To ensure that $f_{\text{probed}}$ emulates the downstream data generating process based on our prior knowledge, we employed the same loss function $\ell(\boldsymbol{\theta}; x, y)$, that is the negative log-likelihood, on the downstream dataset to learn $\mathbf{W}$. For the selection of $\alpha$, we minimize the negative log-likelihood on a held-out validation set which constitutes 10% of the entire training dataset, which has close connections to maximizing the marginal likelihood (Fong & Holmes, 2020).

## 3.2 BLOCK DIRICHLET DISTRIBUTION

Recall the posterior of $F$ after observing the downstream dataset $\mathcal{D} = \{x_i, y_i\}_{i=1}^n$,

$$F | \mathcal{D} \sim \text{DP}\left(\alpha + n, \frac{1}{\alpha + n} \sum_{i=1}^n \delta_{(x_i, y_i)} + \frac{\alpha}{\alpha + n} F_\pi\right). \tag{4}$$

When dealing with a continuous base measure, sampling exactly from the posterior DP is not possible with finite computational resources. Fortunately, the informative base measure $F_\pi$ defined in Equation 3 only involves finite inputs from the downstream dataset. This allows exact sampling using a finite Dirichlet distribution, where we only need to sample weights $(w_{1:n}, \widetilde{w}_{1:n})$ from a Dirichlet distribution defined over $\mathbb{R}_+^{2n}$ with concentration parameters $(1, \dots, 1, \alpha/n, \dots, \alpha/n)$.

Here, $w_{1:n}$ and $\widetilde{w}_{1:n}$ denote the weights associated with the downstream data points and the pseudo data points originating from the base measure $F_\pi$ respectively.

The provided Dirichlet distribution, denoted as $\text{Dir}(1,\ldots,1,\alpha/n,\ldots,\alpha/n)$, generates a weight vector $(w_{1:n},\widetilde{w}_{1:n})$ with a dimension of $2n$. However, due to the numerical issues, accurately generating these weight samples becomes challenging when the number of downstream training data points, denoted as $n$, becomes large. To address this issue, we adopt a block Dirichlet distribution (Shin et al., 2021) to handle the dimensionality of the concentration parameter. To use a block Dirichlet distribution as an alternative to the non-block version, we need to map each element in the set $[n]$ to two pairs of $L$ blocks as we independently block the downstream dataset and the pseudo dataset. This mapping is performed by a function $u : [n] \rightarrow [L] \times [L]$, where $u(i) = (l_1, l_2)$. Here, we denote $u(i)[1]$ as $l_1$ and $u(i)[2]$ as $l_2$. We assign weights $w_i = v_{u(i)[1]}$ for the $i$th training data point and $\widetilde{w}_i$ is assigned as $\tilde{v}_{u(i)[2]}$ for the $i$th pseudo data point, where $(v_{1:L}, \tilde{v}_{1:L}) \sim \text{Dir}(1, \ldots, 1, \alpha/n, \ldots, \alpha/n)$ in $\mathbb{R}_+^{2L}$. After finishing mapping $(w_{1:n}, \tilde{w}_{1:n})$, the sum $\sum_{j=1}^n w_j + \sum_{k=1}^n \widehat{w}_k$ becomes $n/L$ at this point. This index mapping function $u$ is randomly reconstructed for each posterior sample. Please refer to Appendix A for details on the implementation and analysis regarding the asymptotic convergence between the target distributions of the blocked Dirichlet distribution and the non-blocked Dirichlet distribution.

### 3.3 SAMPLING FROM POSTERIOR DISTRIBUTION

To perform posterior sampling, we need to sample a total of $M$ instances of $F^{(m)}$ from the distribution $F|\mathcal{D}$, where $m$ ranges from 1 to $M$. We then need to solve $M$ instances of Equation 2, substituting $F^{(m)}$ for $F_0$. Since our measure has finite support, our objective can be expressed as:

$$\int \ell(\boldsymbol{\theta}; x, y) dF^{(m)}(x, y) = \sum_{j=1}^n w_j^{(m)} \ell(\boldsymbol{\theta}; x_j, y_j) + \sum_{k=1}^n \widetilde{w}_k^{(m)} \ell(\boldsymbol{\theta}; x_k, f_{\text{probed}}(x_k)). \tag{5}$$

As finding the global minima is impractical for deep neural networks and using the entire dataset demands significant memory resources, we have employed Stochastic Gradient Descent (SGD; Robbins & Monro, 1951) methods to optimize the objective described in Equation 5. In order to stabilize the optimization procedure, we maintain the scale of the sum $\sum_{j=1}^n w_j^{(m)} + \sum_{k=1}^n \widetilde{w}_k^{(m)}$ as $2n$ by multiply $2L$ to the variables $(v_{1:L}^{(m)}, \tilde{v}_{1:L}^{(m)})$. This scaling adjustment can be easily achieved by sampling from a distribution, denoted as $(v_{1:L}^{(m)}, \tilde{v}_{1:L}^{(m)}) \sim 2L \times \text{Dir}(1, \ldots, 1, \alpha/n, \ldots, \alpha/n)$. Unlike posterior sampling methods based on MCMC, our approach does not require sequential sampling for each instance, allowing for parallelized posterior sampling. Please refer to Algorithm 1 for a summary of the NPTL algorithm.

## 4 RELATED WORKS

**Posterior sampling in deep neural networks.** While there exist several posterior sampling methods guaranteed by statistical theory, e.g. Hamiltonian Monte Carlo (HMC; Duane et al., 1987; Neal et al., 2011), they are ill-suited for the scale of modern deep neural networks. Consequently, recent advances in Bayesian deep learning have been established upon SGD based methods (Robbins & Monro, 1951), a foundation of modern machine learning algorithms. We can highlight the following research areas for Bayesian deep learning: *1) Variational inference* introduces variational distribution, which are tractable approximate posteriors, and then samples from them (Graves, 2011; Ranganath et al., 2014; Blundell et al., 2015; Khan et al., 2018). *2) Stochastic gradient Markov chain Monte Carlo* directly explores the posterior distribution using stochastic gradients by using various continuous dynamics (Welling & Teh, 2011; Chen et al., 2014; Ma et al., 2015). *3) Particle optimization variational inference* evolves a set of particles towards the high-density regions of the posterior distribution (Liu & Wang, 2016; D'Angelo & Fortuin, 2021). However, most of the investigations into posterior sampling are conducted in *from-scratch* scenarios, where models start with random initializations, and it is considered crucial to specify suitable priors even in the absence of observed data (Fortuin, 2022). Consequently, in transfer learning situations that utilize pre-training data, the significance of establishing priors that consider this becomes even more pronounced.

**Learning priors from data.** Several works have proposed to learn a prior distribution from data. The work by Krishnan et al. (2020) focuses on adapting a prior distribution by maximizing the marginal likelihood using a Maximum Likelihood Estimation solution. On the other hand, Fortuin et al. (2022) takes a different approach by using actual summary statistics of SGD trained weights as a weight prior distribution. Tomczak & Welling (2018) proposed a Variational Mixture of Posteriors (Vamp) prior, which utilizes a mixture of variational posteriors learned from data as a prior. While these approaches aim to create a carefully tailored weight prior distribution based on data, our method establishes an informative nonparametric prior to the true data distribution. This distinction enhances NPTL's resilience for the model misspecification scenarios.

## 5 EXPERIMENTS

In this section, we present empirical evidence that demonstrates the effectiveness of our proposed posterior sampling method in practical transfer learning scenarios. § 5.1 includes experiments conducted on vision tasks, while § 5.2 contains experiments conducted on language tasks. In § 5.3, we suggest a computationally efficient method to decrease inference cost while maintaining the performance of NPTL. Across all the result tables, a **boldfaced underline** highlights the best value, while an underline denotes the second-best value in each column. The last 'Avg.' column summarizes the overall results for each method across all datasets or intensity levels.

As a baseline for assessing the quality of the posterior samples from our proposed NPTL method, we consider the following two representative algorithms for practically implementing BMA; *1) Stochastic Gradient Hamiltonian Monte Carlo (SGHMC):* It is an approximate way to simulate HMC by employing stochastic gradients and is guaranteed to have the same stationary distribution as the original HMC (Chen et al., 2014). Unless otherwise indicated, we utilized 30 posterior samples from the SGHMC sampler for calculating BMA. *2) Ensemble:* Despite its simplicity, it serves as a compelling approach to the practice of BMA (Wilson & Izmailov, 2020). Unless specified, we employed 10 model copies to make ensemble predictions. In the baseline BMA procedure, we employed L2SP (Xuhong et al., 2018) and PTYL (Shwartz-Ziv et al., 2022) approaches, which provide Gaussian priors centered around the pre-trained solution. Appendix B.2 provides a more detailed explanation of these techniques and their specific hyperparameter settings.

### 5.1 EMPIRICAL ANALYSIS ON VISION TASKS

Our experimental setups for visual tasks encompass the following pre-trained models: *1) ResNet-20x4*, a residual network with a depth of 20 and projection shortcuts (He et al., 2016). To better adapt to the transfer learning scenario, we enhanced the model capacity by increasing the number of convolutional filters by four (Zagoruyko & Komodakis, 2016). We then performed supervised pre-training using the downsampled variant of the ImageNet dataset, where the image size is 32x32 (Chrabaszcz et al., 2017). *2) ResNet-50*, a residual network with a depth of 50 and projection shortcuts (He et al., 2016). Fine-tuning the ResNet-50 network, pre-trained on the ImageNet dataset with a 224x224 image size (Russakovsky et al., 2015), is a well-established procedure in computer vision experiments (Girshick et al., 2014; He et al., 2019). *3) ViT-B/16*, a vision transformer (ViT) with $16 \times 16$ input patch size (Dosovitskiy et al., 2021). Considering the recent advancements in computer vision achieved through ViT architectures, we also contemplate scenarios in which we fine-tune the ViT-B/16 network, pre-trained on the ImageNet-21k dataset with a 224x224 image size. We believe the experimental results obtained through these setups will be convincing to the readers.

We verify the effectiveness of our posterior sampling method on a range of downstream image classification tasks when utilizing the aforementioned pre-trained model in a transfer learning context. In addition to measuring classification accuracy (ACC), we also assess the performance of resulting categorical predictions using negative log-likelihood (NLL). NLL provides additional insights into the quality of the posterior predictive distribution obtained through the BMA procedure. Throughout the paper, the values displayed in the tables represent averages with standard deviations calculated over multiple runs. Please refer to Appendix B for a thorough description of downstream datasets, evaluation metrics, and precise hyperparameter information.

**Results for image classification tasks.** We start by evaluating the performance of NPTL on image classification tasks, using the ResNet-20x4 model for the following four datasets with an image size

**Table 1: Main results with BMA for ResNet-20x4.** Evaluation results of NPTL and baseline methods for BMA on four image classification datasets, including `C10`, `C100`, `F101`, and `D120`. Results with standard deviations represent the average of three runs.

| Metric | Methods | Datasets | | | | |
|--------|---------|------|------|------|------|------|
| | | C10 | C100 | F101 | D120 | Avg. |
| ACC (↑) | SGHMC + L2SP | $0.955_{\pm0.000}$ | $0.797_{\pm0.000}$ | $0.625_{\pm0.000}$ | $0.612_{\pm0.001}$ | 0.747 |
| | SGHMC + PTYL | $0.958_{\pm0.001}$ | $0.801_{\pm0.000}$ | $0.632_{\pm0.001}$ | $0.611_{\pm0.008}$ | 0.751 |
| | Ensemble + L2SP | $\underline{0.963}_{\pm0.000}$ | $\underline{0.815}_{\pm0.000}$ | $\mathbf{0.646}_{\pm0.001}$ | $0.624_{\pm0.003}$ | $\underline{0.762}$ |
| | Ensemble + PTYL | $0.958_{\pm0.001}$ | $0.806_{\pm0.000}$ | $0.600_{\pm0.003}$ | $\underline{0.632}_{\pm0.002}$ | 0.749 |
| | NPTL (ours) | $\mathbf{0.964}_{\pm0.001}$ | $\mathbf{0.818}_{\pm0.000}$ | $\underline{0.644}_{\pm0.002}$ | $\mathbf{0.634}_{\pm0.001}$ | $\mathbf{0.765}$ |
| NLL (↓) | SGHMC + L2SP | $0.138_{\pm0.000}$ | $0.686_{\pm0.000}$ | $1.447_{\pm0.001}$ | $1.390_{\pm0.002}$ | 0.915 |
| | SGHMC + PTYL | $0.128_{\pm0.000}$ | $0.665_{\pm0.001}$ | $1.420_{\pm0.000}$ | $1.385_{\pm0.037}$ | 0.900 |
| | Ensemble + L2SP | $\underline{0.107}_{\pm0.001}$ | $\underline{0.617}_{\pm0.001}$ | $\mathbf{1.331}_{\pm0.000}$ | $1.357_{\pm0.000}$ | $\underline{0.853}$ |
| | Ensemble + PTYL | $0.119_{\pm0.000}$ | $0.644_{\pm0.003}$ | $1.513_{\pm0.000}$ | $\underline{1.320}_{\pm0.000}$ | 0.899 |
| | NPTL (ours) | $\mathbf{0.102}_{\pm0.001}$ | $\mathbf{0.606}_{\pm0.002}$ | $\underline{1.347}_{\pm0.001}$ | $\mathbf{1.297}_{\pm0.003}$ | $\mathbf{0.838}$ |

**Table 2: Main results with BMA for ResNet-50.** Evaluation results of NPTL and baseline methods for BMA on five image classification datasets, including `B200`, `C101`, `D47`, `F102`, and `P37`. Results with standard deviations represent the average of three runs.

| Metric | Method | Datasets | | | | | |
|--------|--------|------|------|------|------|------|------|
| | | B200 | C101 | D47 | F102 | P37 | Avg. |
| ACC (↑) | SGHMC + L2SP | $0.782_{\pm0.001}$ | $0.887_{\pm0.001}$ | $0.699_{\pm0.003}$ | $0.903_{\pm0.001}$ | $\mathbf{0.930}_{\pm0.001}$ | 0.840 |
| | SGHMC + PTYL | $0.784_{\pm0.001}$ | $0.884_{\pm0.005}$ | $0.697_{\pm0.004}$ | $0.904_{\pm0.001}$ | $\mathbf{0.930}_{\pm0.002}$ | 0.840 |
| | Ensemble + L2SP | $\underline{0.807}_{\pm0.002}$ | $\underline{0.899}_{\pm0.007}$ | $\underline{0.702}_{\pm0.002}$ | $\mathbf{0.918}_{\pm0.002}$ | $0.924_{\pm0.002}$ | $\underline{0.850}$ |
| | Ensemble + PTYL | $0.805_{\pm0.001}$ | $0.895_{\pm0.005}$ | $\mathbf{0.704}_{\pm0.003}$ | $0.917_{\pm0.000}$ | $0.925_{\pm0.003}$ | 0.849 |
| | NPTL (ours) | $\mathbf{0.811}_{\pm0.003}$ | $\mathbf{0.901}_{\pm0.002}$ | $\underline{0.709}_{\pm0.002}$ | $\underline{0.921}_{\pm0.001}$ | $\mathbf{0.930}_{\pm0.001}$ | $\mathbf{0.854}$ |
| NLL (↓) | SGHMC + L2SP | $0.868_{\pm0.001}$ | $0.350_{\pm0.000}$ | $\underline{1.209}_{\pm0.005}$ | $0.401_{\pm0.003}$ | $0.240_{\pm0.001}$ | 0.614 |
| | SGHMC + PTYL | $0.861_{\pm0.003}$ | $0.358_{\pm0.024}$ | $1.229_{\pm0.005}$ | $0.392_{\pm0.000}$ | $\underline{0.235}_{\pm0.002}$ | 0.615 |
| | Ensemble + L2SP | $0.773_{\pm0.005}$ | $\underline{0.318}_{\pm0.023}$ | $1.295_{\pm0.005}$ | $0.327_{\pm0.002}$ | $0.260_{\pm0.006}$ | 0.595 |
| | Ensemble + PTYL | $\underline{0.773}_{\pm0.003}$ | $0.331_{\pm0.015}$ | $1.288_{\pm0.009}$ | $\underline{0.323}_{\pm0.001}$ | $0.253_{\pm0.004}$ | $\underline{0.594}$ |
| | NPTL (ours) | $\mathbf{0.738}_{\pm0.002}$ | $\mathbf{0.317}_{\pm0.003}$ | $\mathbf{1.167}_{\pm0.002}$ | $\mathbf{0.313}_{\pm0.001}$ | $\mathbf{0.234}_{\pm0.002}$ | $\mathbf{0.554}$ |

of $32 \times 32$: *1) CIFAR-10*, *2) CIFAR-100*, *3) Foods-101*, and *4) Stanford Dogs*. For brevity, these datasets are abbreviated as `C10`, `C100`, `F101`, and `D120`, respectively. Table 1 clearly demonstrates that NPTL outperforms the SGHMC baselines, which serve as a strong competitor in posterior sampling. Although it is widely accepted that ensembles of deep neural networks typically achieve superior performance compared to Bayesian methods (Ashukha et al., 2020; Ovadia et al., 2019), NPTL demonstrates competitive outcomes in terms of ACC and outperforms them in terms of NLL when compared to the ensemble baselines.

We further verify the scalability of our approach through experiments involving the ResNet-50 and ViT-B/16 models. These experiments are conducted on the following five datasets with an image size of $224 \times 224$: *1) Caltech-UCSD Birds 200*, *2) Caltech-101*, *3) Describable Textures Dataset*, *4) Oxford Flowers 102*, and *5) Oxford-IIIT Pet*. To simplify, we use the abbreviations `B200`, `C101`, `D47`, `F102`, and `P37` for these datasets, respectively. The results presented in Table 2 and Table 3 exhibit a consistent trend with the ResNet-20x4 case. It is evident that NPTL demonstrates a significant performance advantage over posterior sampling baselines and surpasses the ensemble baselines. These experimental results highlight that NPTL exhibits superior posterior sampling quality in terms of BMA performance in transfer learning scenarios for image classification tasks.

**Robustness to common corruptions.** We also assess calibration performance by utilizing CIFAR-10-C, a corrupted version of CIFAR-10 (Hendrycks & Dietterich, 2019). This benchmark aims to evaluate the robustness of CIFAR-10 classification models when exposed to 15 common corruptions across five different severity levels. We used posterior samples or ensemble members from the CIFAR-10 classification task to evaluate the performances. Table 4 clearly shows our approach consistently outperforms other baselines across all intensity levels and evaluation metrics we measured. This result shows that posterior samples from NPTL are more robust on distribution shift between the training dataset and the evaluation dataset compared to other baselines. Please refer

**Table 3: Main results with BMA for ViT-B/16.** Evaluation results of NPTL and baseline methods for BMA on five image classification datasets, including `B200`, `C101`, `D47`, `F102`, and `P37`. Results with standard deviations represent the average of three runs.

| Metric | Method | Datasets | | | | | |
|---|---|---|---|---|---|---|---|
| | | B200 | C101 | D47 | F102 | P37 | Avg. |
| ACC (↑) | Ensemble + L2SP | $0.876_{\pm0.002}$ | $0.905_{\pm0.001}$ | $0.765_{\pm0.003}$ | $\mathbf{0.992}_{\pm0.000}$ | $0.939_{\pm0.001}$ | 0.895 |
| | NPTL (ours) | $\mathbf{0.885}_{\pm0.001}$ | $\mathbf{0.923}_{\pm0.001}$ | $\mathbf{0.770}_{\pm0.002}$ | $\mathbf{0.992}_{\pm0.000}$ | $\mathbf{0.944}_{\pm0.000}$ | $\mathbf{0.903}$ |
| NLL (↓) | Ensemble + L2SP | $0.457_{\pm0.001}$ | $0.290_{\pm0.022}$ | $1.005_{\pm0.004}$ | $0.048_{\pm0.000}$ | $0.200_{\pm0.001}$ | 0.400 |
| | NPTL (ours) | $\mathbf{0.420}_{\pm0.002}$ | $\mathbf{0.264}_{\pm0.005}$ | $\mathbf{0.941}_{\pm0.006}$ | $\mathbf{0.049}_{\pm0.001}$ | $\mathbf{0.184}_{\pm0.003}$ | $\mathbf{0.372}$ |

**Table 4: Results on CIFAR-10-C benchmark for ResNet-20x4.** Evaluation results of NPTL and baseline methods for BMA under five different levels of common corruptions. Results with standard deviations represent the average across 15 different corruption types.

| Metrics | Methods | Intensity levels | | | | | |
|---|---|---|---|---|---|---|---|
| | | 1 | 2 | 3 | 4 | 5 | Avg. |
| ACC (↑) | SGHMC + L2SP | $0.844_{\pm0.144}$ | $0.785_{\pm0.176}$ | $0.735_{\pm0.202}$ | $0.667_{\pm0.230}$ | $0.574_{\pm0.236}$ | 0.721 |
| | SGHMC + PTYL | $0.849_{\pm0.144}$ | $0.791_{\pm0.176}$ | $0.740_{\pm0.205}$ | $0.670_{\pm0.236}$ | $0.575_{\pm0.243}$ | 0.725 |
| | Ensemble + L2SP | $\underline{0.875}_{\pm0.116}$ | $\underline{0.823}_{\pm0.150}$ | $\underline{0.778}_{\pm0.180}$ | $\underline{0.713}_{\pm0.211}$ | $\underline{0.625}_{\pm0.220}$ | $\underline{0.763}$ |
| | Ensemble + PTYL | $0.853_{\pm0.137}$ | $0.797_{\pm0.170}$ | $0.749_{\pm0.197}$ | $0.681_{\pm0.229}$ | $0.591_{\pm0.233}$ | 0.734 |
| | NPTL (ours) | $\mathbf{0.890}_{\pm0.098}$ | $\mathbf{0.846}_{\pm0.129}$ | $\mathbf{0.805}_{\pm0.162}$ | $\mathbf{0.750}_{\pm0.191}$ | $\mathbf{0.672}_{\pm0.198}$ | $\mathbf{0.793}$ |
| NLL (↓) | SGHMC + L2SP | $0.475_{\pm0.445}$ | $0.643_{\pm0.531}$ | $0.789_{\pm0.606}$ | $\underline{1.016}_{\pm0.725}$ | $1.339_{\pm0.799}$ | 0.852 |
| | SGHMC + PTYL | $0.465_{\pm0.455}$ | $0.631_{\pm0.542}$ | $\underline{0.777}_{\pm0.619}$ | $1.017_{\pm0.758}$ | $1.345_{\pm0.829}$ | $\underline{0.847}$ |
| | Ensemble + L2SP | $\underline{0.432}_{\pm0.455}$ | $\underline{0.621}_{\pm0.586}$ | $0.798_{\pm0.724}$ | $1.076_{\pm0.889}$ | $1.455_{\pm0.981}$ | 0.876 |
| | Ensemble + PTYL | $0.499_{\pm0.531}$ | $0.703_{\pm0.666}$ | $0.887_{\pm0.790}$ | $1.185_{\pm0.959}$ | $1.556_{\pm1.031}$ | 0.966 |
| | NPTL (ours) | $\mathbf{0.348}_{\pm0.342}$ | $\mathbf{0.500}_{\pm0.456}$ | $\mathbf{0.645}_{\pm0.592}$ | $\mathbf{0.855}_{\pm0.722}$ | $\mathbf{1.142}_{\pm0.787}$ | $\mathbf{0.698}$ |

to Appendix B.3 for more detailed results, including the Expected Calibration Error (ECE; Naeini et al., 2015) metric for measuring the calibration.

## 5.2 EMPIRICAL ANALYSIS ON LANGUAGE TASKS

We further confirm that the applicability of our proposed approach beyond vision models by conducting experiments on language tasks. More precisely, we perform the BMA procedure when fine-tuning the pre-trained *RoBERTa-Base* model (Liu et al., 2019) for the following three subtasks from the GLUE benchmark (Wang et al., 2019): *1) CoLA, 2) MRPC,* and *3) RTE*. These datasets have relatively limited training data (8.55k, 3.67k, and 2.49k, respectively), making the transfer learning process particularly crucial.

Table 5 provides evidence that our proposed NPTL results in a decrease in NLL, implying that it offers an enhanced approach to compute BMA in comparison to the ensemble baseline. In addition to assessing NLL, which measures the quality of the posterior samples used in the BMA process, we also present results using specific evaluation metrics tailored to each dataset within the GLUE benchmark. It is worth noting that our NPTL method also outperforms in terms of GLUE metrics. Here, we have omitted the SGHMC baseline because practical experiments have shown that using SGD with momentum in transformer architectures yields subpar outcomes (Zhang et al., 2020; Liu et al., 2020). However, we believe the *Ensemble* approach would serve as a strong baseline, as demonstrated before in vision experiments. Please refer to Appendix B, for more information regarding datasets, evaluation metrics, and specific hyperparameter details.

## 5.3 SOUPS OF NPL SAMPLES

Despite the theoretical basis of the BMA procedure, a factor that hinders its practicality is the inference cost - the multiple forward passes required for multiple model copies in BMA calculations make applications in resource-constrained real-world deployments challenging. Several methods have been suggested to mitigate this cost issue in ensemble modeling (Izmailov et al., 2018; Malinin et al., 2020; Hobbhahn et al., 2022), and we further introduce NPTL-Soup, a practical variation of NPTL involving weight averaging of posterior samples from NPTL, in line with those attempts.

**Table 5: Main results with BMA for RoBERTa-Base.** Evaluation results of NPTL and baseline methods for BMA on three text classification datasets, including `cola`, `mrpc`, and `rte`. Results with standard deviations represent the average of three runs. †It denotes the use of commonly employed evaluation metrics for each dataset. See Appendix B for further details.

| | | Datasets | | | |
|---|---|---|---|---|---|
| Metric | Method | cola | mrpc | rte | Avg. |
| GLUE (↑)† | Ensemble + L2SP | $0.643_{\pm 0.003}$ | $0.930_{\pm 0.005}$ | $0.794_{\pm 0.006}$ | 0.789 |
| | NPTL (ours) | $\mathbf{0.645}_{\pm 0.006}$ | $\mathbf{0.934}_{\pm 0.003}$ | $\mathbf{0.812}_{\pm 0.006}$ | **0.797** |
| NLL (↓) | Ensemble + L2SP | $0.452_{\pm 0.050}$ | $0.263_{\pm 0.030}$ | $0.518_{\pm 0.049}$ | 0.411 |
| | NPTL (ours) | $\mathbf{0.397}_{\pm 0.021}$ | $\mathbf{0.245}_{\pm 0.009}$ | $\mathbf{0.472}_{\pm 0.006}$ | **0.371** |

**Table 6: Results for soups of NPTL posterior samples.** Evaluation results of NPTL-Soup compared to NPTL. NPTL-Soup requires only one forward pass, resulting in about ten times lower inference cost than the BMA computation of NPTL. †It represents the best value of each metric for each test dataset among the ten fine-tuned models, which can be seen as an unfair advantage given to the 'Fine-tune' baseline.

| | | Datasets | | | | | |
|---|---|---|---|---|---|---|---|
| Metrics | Method | B200 | C101 | D47 | F102 | P37 | Avg. |
| ACC | NPTL | $\mathbf{0.811}_{\pm 0.003}$ | $\mathbf{0.901}_{\pm 0.002}$ | $\mathbf{0.709}_{\pm 0.002}$ | $\mathbf{0.921}_{\pm 0.001}$ | $\mathbf{0.930}_{\pm 0.001}$ | **0.854** |
| | NPTL-Soup | $\underline{0.800}_{\pm 0.007}$ | $\underline{0.894}_{\pm 0.004}$ | $\underline{0.705}_{\pm 0.004}$ | $\underline{0.917}_{\pm 0.001}$ | $\underline{0.923}_{\pm 0.003}$ | $\underline{0.848}$ |
| | Fine-tune | $0.780_{\pm 0.006}$ | $0.879_{\pm 0.009}$ | $0.681_{\pm 0.007}$ | $0.909_{\pm 0.003}$ | $0.906_{\pm 0.006}$ | 0.831 |
| | Fine-tune (oracle)† | 0.785 | 0.893 | 0.695 | 0.912 | 0.921 | 0.841 |
| NLL | NPTL | $\mathbf{0.738}_{\pm 0.002}$ | $\mathbf{0.317}_{\pm 0.003}$ | $\mathbf{1.167}_{\pm 0.002}$ | $\mathbf{0.313}_{\pm 0.001}$ | $\mathbf{0.234}_{\pm 0.002}$ | **0.554** |
| | NPTL-Soup | $\underline{0.755}_{\pm 0.030}$ | $0.357_{\pm 0.031}$ | $\underline{1.329}_{\pm 0.013}$ | $\underline{0.321}_{\pm 0.006}$ | $\underline{0.254}_{\pm 0.009}$ | $\underline{0.603}$ |
| | Fine-tune | $0.881_{\pm 0.019}$ | $0.424_{\pm 0.057}$ | $1.683_{\pm 0.038}$ | $0.358_{\pm 0.016}$ | $0.349_{\pm 0.032}$ | 0.739 |
| | Fine-tune (oracle)† | 0.862 | $\underline{0.340}$ | 1.612 | 0.334 | 0.281 | 0.686 |

Notably, in the transfer learning scenario, fine-tuned solutions from the same initial pre-trained model tend to reside in the same basin on the loss surface (Neyshabur et al., 2020), and we can thus obtain a single solution achieving competitive performance by appropriately averaging weights of NPTL posterior samples. We implemented this NPTL-Soup procedure by employing the Greedy Soup algorithm (Wortsman et al., 2022).

Table 6 demonstrates that NPTL-Soup can achieve performance competitive to that of the BMA computation while reducing the computational demands. Notably, NPTL-Soup achieves superior results compared to the 'Fine-tune (oracle)' baseline, which represents the best-performing fine-tuned solution in terms of *test* performance. While NPTL-Soup does not strictly carry out the BMA procedure, its importance lies in offering a practical *single* solution through the weight-space ensemble of samples with a high posterior density over neural network weights. One could apply alternative methods such as ensemble distillation (Malinin et al., 2020) to mitigate the inherent inference cost associated with the BMA procedure. However, we emphasize that such approaches are not the primary focus of our paper, which primarily introduces the posterior sampling method. We consider them as potential avenues for future research.

## 6 CONCLUSION

In this paper, we proposed a novel posterior sampling approach for transfer learning scenarios, which we call NPTL. NPTL leverages the nonparametric learning (Lyddon et al., 2018; Fong et al., 2019) framework, which effectively accounts for the model misspecification by adapting a nonparametric prior. Our method involves constructing an informative base measure using empirical Bayes techniques and proposing a numerically stable posterior sampling algorithm based on the block Dirichlet distribution. We conducted empirical validations across various tasks and models to validate the performance of NPTL. The results consistently demonstrate that NPTL outperforms other existing methods, indicating its superior ability to produce high-quality posterior samples in transfer learning scenarios.

**Ethics statement.** This paper does not include any ethical issues. This paper proposes a posterior sampling algorithm for the transfer learning scenario which does not cause ethical issues.

**Reproducibility statement.** We described our experimental details in Appendix B which covers dataset description, used libraries, and hardware.

## ACKNOWLEDGEMENT

This work was partly supported by Institute of Information & communications Technology Promotion(IITP) grant funded by the Korea government(MSIT)(No.2019-0-00075, Artificial Intelligence Graduate School Program(KAIST)), the National Research Foundation of Korea(NRF) grant funded by the Korea government(MSIT) (NRF-2022R1A5A708390812, NRF-2021M3E5D9025030), and KAIST-NAVER Hypercreative AI Center.

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

## A    DISCUSSION ON BLOCK DIRICHLET DISTRIBUTION

### A.1    IMPLEMENTATION

Let $I_1, \ldots, I_L$ denote the randomly constructed pairwise disjoint index sets where $\bigcup_{i=1}^{L} I_i = [n]$ where $|I_i| = \frac{n}{L}$ for all $i \in [L]$ and $n$ is the number of the training dataset. Furthermore, let $I_1', \ldots, I_L'$ denote another randomly constructed pairwise disjoint index sets where $\bigcup_{i=1}^{L} I_i' = [n]$ where $|I_i'| = \frac{n}{L}$ for all $i \in [L]$. Now let index mapping functions $u : [n] \to [L] \times [L]$ where $u(i) = (l_1, l_2)$ if $i \in I_{l_1}$ and $i \in I_{l_2}$. Here we denote $u(i)[1] = l_1$ and $u(i)[2] = l_2$. Then, given some scaled Dirichlet weights $(v_1, ..., v_n, \tilde{v}_1, ..., \tilde{v}_L) \sim 2L \times \text{Dir}(1, ..., 1, \alpha/n, ..., \alpha/n)$ on $\mathbb{R}_+^{2L}$, we assign the weights for the $i$th training datum and $i$-th pseudo datum as $w_i = v_{u(i)[1]}$ and $\widetilde{w}_i = \tilde{v}_{u(i)[2]}$ .

### A.2    CONVERGENCE

In this section, we will show that a simplified version of the blocked posterior bootstrap is asymptotically equivalent to the non-blocked bootstraps based on the assumptions in Theorem A.1 of Shin et al. (2021). The simplification involves keeping $T$ fixed and not blocking the pseudo data. We leave the extension of our theory to the implemented case for future work. Formally, let $I_1, \ldots, I_L$ denotes the randomly constructed pairwise disjoint index sets where $\bigcup_{i=1}^{L} I_i = [n]$ and $n$ is the size of the training dataset, and again $|I_i| = \frac{n}{L}$ for all $i \in [L]$. Now define the index mapping functions $u : [n] \to [L]$ where $u(i) = l$ if $i \in I_l$. Here we denote $u(i) = l$. Then, given some Dirichlet weights $(v_1, ..., V_n, \tilde{v}_1, ..., \tilde{v}_T) \sim \text{Dir}(1, ..., 1, \alpha/T, ..., \alpha/T)$ on $\mathbb{R}_+^{L+T}$, we assign the weight $w_i = v_{u(i)}$ for the $i$-th training datum and $\widetilde{w}_j = \tilde{v}_j$ for the $j$-th pseudo datum. We have omitted the scaling term for the theoretical study, and assume that $L$ is a fixed constant.

We assume that the data points $Z_1, Z_2, \ldots$ are i.i.d. from the probability measure $\mathbb{P}_0$, where we write $Z_i = (X_i, Y_i)$. We then denote the empirical probability measure for the $n$ observations by $\mathbb{P}_n := \frac{1}{n} \sum_{i=1}^{n} \delta_{Z_i}$. For any probability measure $\mathbb{P}$ and a $\mathbb{P}$-measurable function $g$, let $\mathbb{P}g$ denote $\int g \, d\mathbb{P}$. The non-blocked weighted posterior distribution can be written as

$$F = \sum_{i=1}^{n} \zeta_i \delta_{Z_i} + \sum_{j=1}^{T} \widetilde{\zeta}_j \delta_{\widetilde{Z}_j}$$

where $\widetilde{Z}_j \overset{\text{i.i.d.}}{\sim} F_\pi$ and $\{\zeta_1, \ldots, \zeta_n, \widetilde{\zeta}_1, \ldots, \widetilde{\zeta}_T\} \sim \text{Dir}(1, \ldots, 1, \alpha/T, \ldots, \alpha/T)$ independently. Here, we assume $\alpha$ and $T$ are constant and $F_\pi$ is a fixed distribution from which we draw $T$ samples. We will also write $\hat{F}_\pi = \frac{1}{T} \sum_{j=1}^{T} \delta_{\widetilde{Z}_j}$. The blocked weighted posterior distribution is then

$$F_B = \frac{1}{\sum_{i=1}^{n} w_i + \sum_{j=1}^{T} \tilde{w}_j} \left[ \sum_{i=1}^{n} w_i \delta_{Z_i} + \sum_{j=1}^{T} \widetilde{w}_j \delta_{\widetilde{Z}_j} \right]$$

where $\{w_1, \ldots, w_n, \widetilde{w}_1, \ldots, \widetilde{w}_T\}$ is described just above. Note that we have

$$\sum_{i=1}^{n} w_i + \sum_{j=1}^{T} \widetilde{w}_j = 1 + \left( \frac{n}{L} - 1 \right) \sum_{i=1}^{L} v_i.$$

We will now show that $F$ and $F_B$ are asymptotically equivalent.

First, we study $F$ following Theorem 12.2 from Ghosal & Van der Vaart (2017).

**Proposition A.1** (Ghosal & Van der Vaart (2017)). *Let $\mathcal{H}$ be class of functions that is $\mathbb{P}_0$-Donsker, with envelope function $H(z) = \sup_{h \in \mathcal{H}} |h(z)|$ which satisfies both $\mathbb{P}_0^*[H^2] < \infty$ and $\hat{F}_\pi^*[H^2] < \infty$. The process $\sqrt{n}(F - \mathbb{P}_n)$, where $F$ is the non-blocked weighted posterior, converges conditionally in distribution given $Z_1, Z_2, \ldots$ in $L_\infty(\mathcal{F})$ to $\mathbb{G}$ as $n \to \infty$ $\mathbb{P}_0^\infty$-a.s., where $\mathbb{G}$ is a Brownian bridge process.*

*Proof.* First, we note from Proposition G.10 of Ghosal & Van der Vaart (2017) that $F$ can be written as

$$F = V_n Q + (1 - V_n)\mathbb{B}_n$$

where $V_n \sim \text{Be}(\alpha, n)$, $\mathbb{B}_n \sim \text{DP}(n, \mathbb{P}_n)$ and $Q \sim \text{DP}\left(\alpha, \hat{F}_\pi\right)$ are independent on an appropriate product probability space. Here, $\text{Be}(\alpha, \beta)$ denotes the beta distribution with parameters $\alpha$ and $\beta$. The process can then be written as

$$\sqrt{n}(F - \mathbb{P}_n) = \sqrt{n}V_n (Q - \mathbb{B}_n) + \sqrt{n}(\mathbb{B}_n - \mathbb{P}_n).$$

For the first term on the right, $\sqrt{n}V_n \to 0$ in probability, and $\mathbb{B}_n$ converges to a limit in $L_\infty(\mathcal{F})$. As the mapping $h \to Qh$ is in $L_\infty(\mathcal{F})$, the process $\sqrt{n}V_n (Q - \mathbb{B}_n)$ converges to 0 in probability in $L_\infty(\mathcal{F})$.

The second term $\sqrt{n}(\mathbb{B}_n - \mathbb{P}_n)$ is well understood and converges conditionally in distribution given $Z_1, Z_2, \ldots$ to $\mathbb{G}$ in $L_\infty(\mathcal{F})$ $\mathbb{P}_0^\infty$-a.s., and applying Slutsky's theorem gives the result. $\square$

This Proposition A.1 shows that $\sqrt{n}(F - \mathbb{P}_n)$ converges weakly to $\mathbb{G}$ when $n \to \infty$. Now, we study $F_B$ with a similar process in Proposition A.1, following Shin et al. (2021).

**Proposition A.2.** *Let $\mathcal{H}$ be class of functions that is $\mathbb{P}_0$-Donsker, with envelope function $H(z) = \sup_{h \in \mathcal{H}} |h(z)|$ which satisfies both $\mathbb{P}_0^*[H^2] < \infty$ and $\hat{F}_\pi^*[H^2] < \infty$. Assume that $L \to \infty$ as $n \to \infty$. Then the process $\sqrt{n}(F_B - \mathbb{P}_n)$, where $F_B$ is the blocked weighted posterior, converges conditionally in distribution given $Z_1, Z_2, \ldots$ in $L_\infty(\mathcal{F})$ to $\mathbb{G}$ as $n \to \infty$ $\mathbb{P}_0^\infty$-a.s., where $\mathbb{G}$ is a Brownian bridge process.*

*Proof.* Note that $F_B$ can be written as

$$F_B = V_n Q + (1 - V_n)\mathbb{B}_n$$

where $Q \sim \text{DP}\left(\alpha, \hat{F}_\pi\right)$ as before, and

$$\mathbb{B}_n = \frac{1}{n}\sum_{i=1}^{n}\eta_{u(i)}\delta_{Z_i}$$

where $(\eta_1, \ldots, \eta_L) \sim L \times \text{Dir}(1, \ldots, 1)$. Finally, $V_n \sim G/(G + H_n)$ where we have independently $G \sim \text{Ga}(\alpha, 1)$, $H_n \sim \text{Ga}(L, L/n)$. As before, $V_n$, $Q$ and $\mathbb{B}_n$ are all independent on an appropriate product probability space.

To see this, we note that $F_B$ can be written as

$$F_B = \frac{1}{1 + \left(\frac{n}{L} - 1\right)\sum_{i=1}^{L}\frac{\gamma_i}{\gamma + \widetilde{\gamma}}} \times \frac{1}{\gamma + \widetilde{\gamma}}\left[\sum_{j=1}^{T}\widetilde{\gamma}_j\,\delta_{\widetilde{Z}_j} + \sum_{i=1}^{n}\gamma_{u(i)}\,\delta_{Z_i}\right]$$

$$= \frac{1}{\widetilde{\gamma} + \frac{n}{L}\gamma}\left[\sum_{j=1}^{T}\widetilde{\gamma}_j\,\delta_{\widetilde{Z}_j} + \sum_{i=1}^{n}\gamma_{u(i)}\,\delta_{Z_i}\right]$$

where $\widetilde{\gamma}_{1:T} \overset{\text{i.i.d.}}{\sim} \text{Ga}(\alpha/T, 1)$, $\gamma_{1:L} \overset{\text{i.i.d.}}{\sim} \text{Ga}(1, 1)$ and $\gamma = \sum_{l=1}^{L}\gamma_l$ and $\widetilde{\gamma} = \sum_{j=1}^{T}\widetilde{\gamma}_j$. Here, $\text{Ga}(\alpha, \beta)$ denotes the Gamma distribution with parameters $\alpha$ and $\beta$. This follows from the Gamma construction of the Dirichlet distribution. Following the proof of Proposition G.10 of Ghosal & Van der Vaart (2017), define

$$V_n = \frac{\widetilde{\gamma}}{\widetilde{\gamma} + \frac{n}{L}\gamma}, \quad Q = \sum_{j=1}^{T}\frac{\widetilde{\gamma}_j}{\widetilde{\gamma}}\delta_{\widetilde{Z}_j}, \quad \mathbb{B}_n = \frac{L}{n}\sum_{i=1}^{n}\frac{\gamma_{u(i)}}{\gamma}\delta_{Z_i}.$$

It is clear that the above has the correct marginal distributions. From Proposition G.2(i) of Ghosal & Van der Vaart (2017), we note that $Q$ is independent of the normalizing constant $\widetilde{\gamma}$ (and obviously of

$\gamma_{1:L}$ and $\gamma$). Likewise, $\mathbb{B}_n$ is independent of $\gamma$ (and of $\widetilde{\gamma}_{1:T}, \widetilde{\gamma}$), so $V_n$, $Q$ and $\mathbb{B}_n$ are all independent. Finally, we have

$$V_n Q = \frac{1}{\widetilde{\gamma} + \frac{n}{L}\gamma} \sum_{j=1}^{T} \widetilde{\gamma}_j \, \delta_{\widetilde{Z}_j}$$

and similarly

$$(1 - V_n)\mathbb{B}_n = \frac{1}{\widetilde{\gamma} + \frac{n}{L}\gamma} \sum_{i=1}^{n} \gamma_{u(i)} \delta_{Z_i}$$

which together gives $F_B = V_n Q + (1 - V_n)\mathbb{B}_n$.

The process under study can then be written as

$$\sqrt{n}(F_B - \mathbb{P}_n) = \sqrt{n}V_n \left( Q - \widetilde{\mathbb{B}}_n \right) + \sqrt{n} \left( \widetilde{\mathbb{B}}_n - \mathbb{P}_n \right).$$

The second term $\sqrt{n} \left( \widetilde{\mathbb{B}}_n - \mathbb{P}_n \right)$ is shown to converge conditionally in distribution given $Z_1, Z_2, \ldots$ to $\mathbb{G}$ in $L_\infty(\mathcal{F})$ $\mathbb{P}_0^\infty$-a.s. in Theorem A.1 of Shin et al. (2021) under our assumptions. This relies on the fact that $\{\eta_{u(1)}, \ldots, \eta_{u(n)}\}$ are exchangeable due to the random allocation of data points into groups and the result of Præstgaard & Wellner (1993).

For the first term on the right, we have that $\sqrt{n}V_n \to 0$ in probability. To see this, note that

$$E[V_n] = E \left[ \frac{G}{G + H_n} \right] \le E \left[ \frac{G}{H_n} \right]$$
$$= \frac{E[G]}{E[H_n]} = \frac{\alpha}{n}.$$

As a result, $\sqrt{n}E[V_n] \to 0$, and from Markov's inequality we have $\sqrt{n}V_n \xrightarrow{p} 0$. As before, this gives $\sqrt{n}V_n \left( \widetilde{Q} - \mathbb{B}_n \right) \to 0$ in probability in $L_\infty(\mathcal{F})$, which gives the desired result. $\square$

As $\sqrt{n}(F - \mathbb{P}_n)$ and $\sqrt{n}(F_B - \mathbb{P}_n)$ converge to the same limit, we can also apply the rest of Theorem A.1 of Shin et al. (2021) under appropriate assumptions. This shows that the target distribution of a simplified version of the blocked posterior bootstrap is asymptotically equivalent to the target distribution of non-blocked bootstrap.

## B  SUPPLEMENTARY MATERIAL ON EXPERIMENTS

Our code is constructed using the following libraries, which are available under the Apache-2.0 licence[1]: JAX (Bradbury et al., 2018), Flax (Babuschkin et al., 2020), Optax (Babuschkin et al., 2020), TensorFlow Datasets (Abadi et al., 2015), and Transformers (Wolf et al., 2020). We intend to make the code available to the public once the research has been published. All experiments were conducted on NVIDIA RTX 3090 GPU machines.

### B.1  EXPERIMENTAL DETAILS

**Datasets for ResNet-20x4 experiments.** The ResNet-20x4 experiments make use of the following image classification datasets. The network processes input images with dimensions of $32 \times 32 \times 3$, and all the images are standardized through the subtraction of $(0.481, 0.458, 0.408)$ and division by $(0.269, 0.261, 0.276)$.

- `C10` and `C100` : CIFAR-10/100[2] (Krizhevsky et al., 2009) consists of 10/100 classes sourced from 80 Million Tiny Images (Torralba et al., 2008), and every image in this dataset has sizes of $32 \times 32$. We allocated the 60,000 images publicly available into splits of 45,000 for training, 5,000 for validation, and 10,000 for testing.

---

[1]https://www.apache.org/licenses/LICENSE-2.0
[2]https://www.cs.toronto.edu/~kriz/cifar.html

- `F101` : Food-101[3] (Bossard et al., 2014) originally consists of 101 food categories, and the images in this dataset have a maximum side length of 512 pixels. We center-cropped the images, resized them to $32 \times 32$, and divided the 101,000 publicly available images into 61,440 for training, 14,310 for validation, and 25,250 for testing.

- `D120` : Stanford Dogs[4] (Khosla et al., 2011) comprises pictures of 120 dogs breeds sourced from ImageNet (Russakovsky et al., 2015). We conducted center-cropping on these images, resized them to $32 \times 32$, and partitioned the publicly available set of 20,580 images into 10,240 for training, 1,760 for validation, and 8,580 for testing.

**Datasets for ResNet-50 and ViT-B/16 experiments.** The following image classification datasets are employed in the ResNet-50 and ViT-B/16 experiments. Both neural network architectures handle input images with dimensions of $224 \times 224 \times 3$, and all these images are standardized by subtracting $(0.481, 0.458, 0.408)$ and dividing by $(0.269, 0.261, 0.276)$.

- `B200` : Caltech-UCSD Birds 200[5] (Wah et al., 2011) comprises pictures of 200 bird species, where some images overlap with images in ImageNet (Russakovsky et al., 2015). We allocated the 11,788 images publicly available into split of 5,394 for training, 300 for validation, and 5,994 for testing.

- `C101` : Caltech-101[6] (Li et al., 2022) comprises 101 categories, where the images generally have edge lengths ranging from 200 to 300 pixels. We divided the 9,144 publicly available images into subsets of 2,754 for training, 306 for validation, and 6,084 for testing.

- `D47` : Describable Textures Dataset[7] (Cimpoi et al., 2014) consists of 47 categories, and the image sizes range between $300 \times 300$ and $640 \times 640$. We utilized a total of 5,640 images, with 1,880 for training, 1,880 for validation, and 1,880 for testing.

- `F102` : Oxford Flowers 102[8] (Nilsback & Zisserman, 2008) comprises 102 distinct flower categories commonly found in the United Kingdom. We used a dataset consisting of a total of 8,189 images, splitted as 1,020 for training, 1,020 for validation, and 6,149 for testing.

- `P37` : Oxford-IIIT Pet[9] (Parkhi et al., 2012) comprises 37 categories, where all images have an associated ground truth annotation of breed. We divided the 7,349 images publicly available into split of 3,312 for training, 368 for validation, and 3,669 for testing.

**Datasets for RoBERTa-Base.** The RoBERTa-Base experiments utilize the following subtasks from the GLUE benchmark. The network processes token sequences with a maximum length of 512 tokens, as per the default setup described in Liu et al. (2019). In the main text, we reported the performance on the validation split.

- `cola` : Corpus of Linguistic Acceptability (Warstadt et al., 2018) comprises 8.55k sentences designated for training and 1.04k sentences designated for validation.

- `mrpc` : Microsoft Research Paraphrase Corpus (Dolan & Brockett, 2005) includes 3.67k pairs of sentences for training and 408 pairs of sentences for validation.

- `rte` : Recognizing Textual Entailment (RTE) originated from a sequence of challenges for textual entailment, including RTE1 (Dagan et al., 2006), RTE2 (Bar Haim et al., 2006), RTE3 (Giampiccolo et al., 2007), and RTE5 (Bentivogli et al., 2009). Consequently, it comprises 2.49k pairs of sentences for training and 277 pairs of sentences for validation.

**Architectural details on ResNet-20x4.** In line with Izmailov et al. (2021), we endeavored in our ResNet-20x4 experiments to establish a clear Bayesian interpretation through the following experimental configuration; 1) We opted not to utilize any data augmentation techniques. 2) We employed Swish activation (Hendrycks & Gimpel, 2016; Elfwing et al., 2018; Ramachandran et al., 2017) to

---

[3] https://data.vision.ee.ethz.ch/cvl/datasets_extra/food-101/

[4] http://vision.stanford.edu/aditya86/ImageNetDogs/main.html

[5] http://www.vision.caltech.edu/datasets/cub_200_2011/

[6] https://data.caltech.edu/records/mzrjq-6wc02

[7] https://www.robots.ox.ac.uk/~vgg/data/dtd/index.html

[8] https://www.robots.ox.ac.uk/~vgg/data/flowers/102/

[9] https://www.robots.ox.ac.uk/~vgg/data/pets/

**Table 7: Hyperparameter setup.** It summarizes the hyperparameter settings employed in the experimental results presented in the main text.

**(a) ResNet-20x4**

|      | B200 | C101 | D47 | F102 |
|------|------|------|-----|------|
| L2SP | $\beta = 10^{-7}$ | $\beta = 10^{-7}$ | $\beta = 10^{-7}$ | $\beta = 10^{-10}$ |
| PTYL | $\gamma = 10^{16}$ | $\gamma = 10^{16}$ | $\gamma = 10^{16}$ | $\gamma = 10^{2}$ |
| NPTL | $\alpha = 1.0$ | $\alpha = 1.0$ | $\alpha = 1.0$ | $\alpha = 1.0$ |

**(b) ResNet-50 and ViT-B/16**

|           |      | B200 | C101 | D47 | F102 | P37 |
|-----------|------|------|------|-----|------|-----|
| ResNet-50 | L2SP | $\beta = 10^{-3}$ | $\beta = 10^{-3}$ | $\beta = 10^{-4}$ | $\beta = 10^{-4}$ | $\beta = 10^{-4}$ |
|           | PTYL | $\gamma = 10^{12}$ | $\gamma = 10^{12}$ | $\gamma = 10^{12}$ | $\gamma = 10^{12}$ | $\gamma = 10^{12}$ |
|           | NPTL | $\alpha = 10^{3}$ | $\alpha = 10^{0}$ | $\alpha = 10^{2}$ | $\alpha = 10^{2}$ | $\alpha = 10^{2}$ |
| ViT-B/16  | L2SP | $\beta = 10^{-4}$ | $\beta = 10^{-4}$ | $\beta = 10^{-4}$ | $\beta = 10^{-4}$ | $\beta = 10^{-4}$ |
|           | NPTL | $\alpha = 1.0$ | $\alpha = 1.0$ | $\alpha = 1.0$ | $\alpha = 1.0$ | $\alpha = 1.0$ |

**(c) RoBERTa-Base**

|      | cola | mrpc | rte |
|------|------|------|-----|
| L2SP | $\beta = 10^{-2}$ | $\beta = 10^{-2}$ | $\beta = 10^{-2}$ |
| NPTL | $\alpha = 10^{-2}$ | $\alpha = 10^{-2}$ | $\alpha = 10^{-2}$ |

attain smoother parameter posteriors. 3) We replaced batch normalization layers (Ioffe & Szegedy, 2015) with filter response normalization layers (Singh & Krishnan, 2020). Although these setups might result in reduced performance compared to conventional experimental configurations (such as training the network with ReLU activations, batch normalization layers, and data augmentation strategies), we believe that these setups are likely to produce results that are clearer from a Bayesian perspective. It is worth noting that our ResNet-50 experiments address the conventional configurations with ReLU activation and batch normalization layers.

**Optimization details on ResNet-20x4 and ResNet-50.** We utilized an SGD optimizer with momentum (Polyak, 1964). The momentum value was kept constant at 0.9, and we experimented with different base learning rates using the cosine decay schedule, specifically testing values in the range of {0.1, 0.03, 0.01}. For the ResNet-20x4 experiments, training terminated after 10 epochs with a batch size of 80. For the ResNet-50 experiments, optimization halted after 5,000 training steps with a batch size of 128. These experiments were conducted on a single GPU machine. During the SGHMC experiments, we implemented posterior tempering as a strategy to mitigate the impact of the *cold posterior effect* (Wenzel et al., 2020), employing a posterior temperature setting of 0.0001.

**Optimization details on ViT-B/16 and RoBERTa-Base.** For the ViT-B/16 and RoBERTa-Base models, we employed an Adam optimizer (Kingma & Ba, 2015). The momentum hyperparameters were set to their default value from the Optax library, which are 0.9 for the first moment and 0.999 for the second moment. We configured the base learning rates at 0.0001 for ViT-B/16 and 0.00003 for RoBERTa-Base. In the case of ViT-B/16 experiments, we applied a cosine decay schedule, while for RoBERTa-Base experiments, a linear decay schedule with warmup was used. The optimization terminated after 5,000 training steps with a batch size of 128 for ViT-B/16 experiments conducted on distributed training with two GPU machines. For RoBERTa-Base experiments, the optimization terminated after 10 training epochs with a batch size of 16 on a single GPU machine.

## B.2 METHODS AND HYPERPARAMETERS

Table 7 summarizes a comprehensive outline of the hyperparameter settings for each experiment.

**Table 8: Calibration results for CIFAR-10-C.** Expected calibration error (ECE) of NPTL and baseline methods for BMA under five different levels of common corruptions. Results with standard deviations represent the average across 15 different corruption types. A **boldfaced underline** highlights the best value, while an underline denotes the top two values in each column. The last 'Avg.' column summarizes the overall results for each method across all intensity levels. It supplements Table 4 presented in the main text.

| Metrics | Methods | Intensity levels | | | | | Avg. |
|---|---|---|---|---|---|---|---|
| | | 1 | 2 | 3 | 4 | 5 | |
| ECE ($\downarrow$) | SGHMC + L2SP | $0.039_{\pm 0.020}$ | $0.050_{\pm 0.043}$ | $0.060_{\pm 0.071}$ | $0.092_{\pm 0.093}$ | $0.138_{\pm 0.134}$ | 0.076 |
| | SGHMC + PTYL | $0.036_{\pm 0.027}$ | $0.048_{\pm 0.048}$ | $0.063_{\pm 0.081}$ | $0.098_{\pm 0.110}$ | $0.149_{\pm 0.144}$ | 0.079 |
| | Ensemble + L2SP | $0.046_{\pm 0.057}$ | $0.071_{\pm 0.078}$ | $0.096_{\pm 0.102}$ | $0.135_{\pm 0.123}$ | $0.191_{\pm 0.142}$ | 0.108 |
| | Ensemble + PTYL | $0.051_{\pm 0.067}$ | $0.078_{\pm 0.089}$ | $0.104_{\pm 0.111}$ | $0.145_{\pm 0.133}$ | $0.200_{\pm 0.145}$ | 0.116 |
| | NPTL (ours) | **$0.025_{\pm 0.036}$** | **$0.040_{\pm 0.053}$** | **$0.058_{\pm 0.076}$** | **$0.084_{\pm 0.094}$** | **$0.123_{\pm 0.108}$** | **0.066** |

**A strength of prior belief in NPTL.** NPTL includes a hyperparameter $\alpha$ that signifies the strength of the prior belief. We summarize a detailed hyperparameter configuration for each experiment:

- In ResNet-20x4 and ResNet-50 experiments, we swept over the logarithmically spaces values of $\alpha \in \{10^k\}_k$.

- In ViT-B/16 experiments, we set a constant value of $\alpha = 1.0$ without any sweeping.

- In RoBERTa-Base experiments, we set $\alpha$ to a constant value of $0.01$ after exploring different values of $\alpha \in \{1.0, 0.1, 0.01\}$.

**A prior variance in L2SP and PTYL.** Consider a Gaussian prior over neural network parameters $\phi$ denoted as $p(\phi) = \mathcal{N}(\phi; \mu, \Sigma)$, where $\mu$ represents the mean and $\Sigma$ signifies the covariance. (1) The L2SP prior corresponds to a scenario where $\mu = \phi^*$ and $\Sigma = \beta \mathbf{I}$. Here, $\phi^*$ denotes the pre-trained parameters, and $\beta$ serves as a hyperparameter responsible for adjusting the prior variance. (2) The PTYL prior corresponds to a scenario where $\mu = \phi_{\text{SWA}}^*$ and $\Sigma = \gamma \Sigma_{\text{SWAG}}^*$. Here, $\phi_{\text{SWA}}^*$ and $\Sigma_{\text{SWAG}}^*$ are determined through the SWAG procedure using upstream data, while $\gamma$ acts as a hyperparameter controlling the scale of prior variance. Following the official implementation code of Shwartz-Ziv et al. (2022)[10], we introduced an additional hyperparameter $\varepsilon = 0.1$ to avoid any numerical issues by defining the covariance as $\Sigma = \gamma \Sigma_{\text{SWAG}}^* + \varepsilon \mathbf{I}$. We summarize a detailed hyperparameter configuration for each experiment:

- In ResNet-20x4 experiments, we swept over the logarithmically spaced values of $1/\beta \in \{2.5 \times 2^k\}_k$ and $\gamma \in \{2^k\}_k$.

- In ResNet-50 experiments, we instead tuned $1/(\beta|\mathcal{D}|)$ to align with the established practical weight decay regularization convention, where $|\mathcal{D}|$ denotes the number of training examples. We swept over the logarithmically spaced values of $1/(\beta|D|) \in \{10^{-k}\}_k$ and $\gamma \in \{10^k\}_k$.

- We fixed $1/(\beta|\mathcal{D}|) = 0.0001$ in ViT-B/16 experiments and $1/(\beta|\mathcal{D}|) = 0.01$ in RoBERTa-Base experiments.

### B.3 SUPPLEMENTARY RESULTS FOR CIFAR-10-C BENCHMARK

**Measuring calibration error.** Table 8 further provides the evaluation results using ECE. Again, NPTL surpasses other baselines, indicating it provides well-calibrated predictions.

**Results in box plots.** Figure 1 visually presents supplementary statistics regarding the results, including the median and the first and third quartiles, depicted through a box plot. Overall, NPTL achieves lower NLL and ECE.

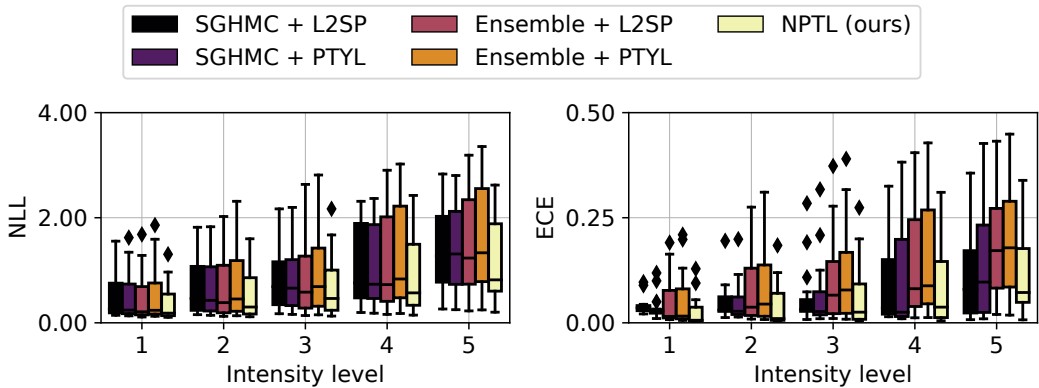

Figure 1: **Box plots for CIFAR-10-C.** It supplements Table 4 presented in the main text.

Table 9: **Additional BMA results for ResNet-20x4.** Evaluation results of NPTL and baseline method for BMA on four image classification datasets, including `C10`, `C100`, `F101`, and `D120`. Results with standard deviations represent the average of three runs. Here, IG implies the zero-mean isotropic Gaussian prior.

| Metric | Methods | Datasets | | | | |
|---|---|---|---|---|---|---|
| | | C10 | C100 | F101 | D120 | Avg. |
| ACC ($\uparrow$) | SGHMC + IG | $0.872_{\pm0.000}$ | $0.625_{\pm0.001}$ | $0.550_{\pm0.001}$ | $0.235_{\pm0.001}$ | 0.571 |
| | NPTL (ours) | $0.964_{\pm0.001}$ | $0.818_{\pm0.000}$ | $0.644_{\pm0.002}$ | $0.634_{\pm0.001}$ | 0.765 |
| NLL ($\downarrow$) | SGHMC + IG | $0.381_{\pm0.000}$ | $1.321_{\pm0.001}$ | $1.733_{\pm0.003}$ | $3.071_{\pm0.002}$ | 1.627 |
| | NPTL (ours) | $0.102_{\pm0.001}$ | $0.606_{\pm0.002}$ | $1.347_{\pm0.001}$ | $1.297_{\pm0.003}$ | 0.838 |

### B.4 SUPPLEMENTARY RESULTS FOR RESNET-20X4

If the chosen prior distribution is not appropriate, it can have a substantial impact on the performance of BMA with posterior samples. To exemplify this, we present the BMA performance using the zero-mean isotropic Gaussian prior, denoted as $\mathcal{N}(\mathbf{0}, \sigma^2 * I)$. The empirical results in Table 9 clearly demonstrate how an ill-suited prior distribution can significantly deteriorate performance.

### B.5 SUPPLEMENTARY RESULTS FOR RESNET-50

**Ablation study on Dirichlet concentration.** Here, we conducted an ablation study on concentration parameter $\alpha$. As we already mentioned in § 3.1, $\alpha$ indicates the amount of our belief in prior knowledge. Figure 2 shows that when distribution shifts occur between the upstream dataset and the downstream dataset (s.a. `C101`, `D47`), the gradual increment of alpha leads to a decline in the model's performance. However, if the upstream dataset and the downstream dataset are similar (s.a. `F102`, `P37`), increasing $\alpha$ value leads to enhanced model performance.

**Ablation study on block Dirichlet.** When the number of downstream training data $n$ increases, sampling accurate values from Dirichlet distribution becomes challenging due to the numerical issues. To empirically validate this point, we conducted an experiment for the image classification task. Specifically, we confirmed a significant decline in ACC and a rise in NLL when we replaced block Dirichlet (where the number of blocks is 10) with non-block Dirichlet in the ResNet-50 experiments conducted on `C101`. The test ACC decreased from $0.901_{\pm0.002}$ to $0.880_{\pm0.001}$, while the NLL increased from $0.317_{\pm0.003}$ to $0.375_{\pm0.002}$. Hinging on this observation, we fixed the number of blocks to 10 throughout the paper.

---

[10]https://github.com/hsouri/BayesianTransferLearning

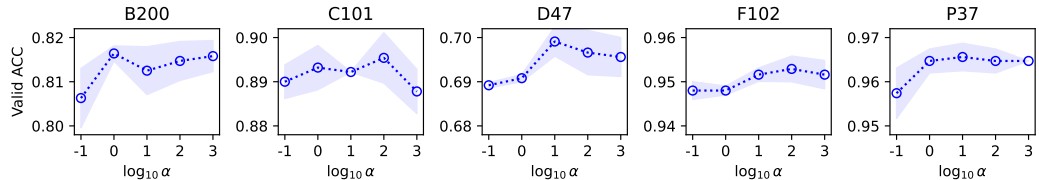

**Figure 2: Ablations on prior belief in NPTL.** It highlights that the optimal $\alpha$ value, representing the prior belief towards the pre-trained model, varies across diverse downstream datasets.

**Table 10: Mean IoU results for PASCAL VOC 2012.** BMA Mean IoU performance of SGLD with L2SP, SGLD with supervised pre-trained PTYL, SGLD with self-supervised pre-trained PTYL, and NPTL with ResNet-50 backbone.

| Dataset | Backbone | L2SP | SL PTYL | SSL PTYL | NPTL |
|---------|----------|------|---------|----------|------|
| PASCAL VOC 2012 | ResNet-50 | 73.16 | 73.72 | 74.15 | 74.72 |

### B.6 SUPPLEMENTARY RESULTS FOR SEMANTIC SEGMENTATION EXPERIMENTS

In this subsection, we extended our experiments to semantic segmentation using the PASCAL VOC 2012 dataset, following the approach of Shwartz-Ziv et al. (2022). Utilizing the pre-trained ResNet-50 as our backbone model, we employed the DeepLabv3+ (Chen et al., 2017) model as our complete model. Our experimental setup, except for our model, remained consistent with that of Shwartz-Ziv et al. (2022). Table 10 presents conclusive evidence that NPTL outperforms other baseline models.

## C DISCUSSION ON THE LIMITATION OF THE NPTL APPROACH

The potential drawback of the suggested NPTL approach lies in the additional training cost. Specifically, implementing the proposed NPTL algorithm incurs extra computational expenses for obtaining $f_{\text{probed}}$ through linear probing on the given downstream data. However, it is important to note that the computational cost associated with linear probing is relatively minimal compared to the training of the entire network.

## D DISCUSSION ON ROBUSTNESS OF THE NPL POSTERIOR TO MODEL MISSPECIFICATION

In this section, we explore the robustness of the NPL posterior compared to the regular Bayesian posterior on $\boldsymbol{\theta}$ in the face of model misspecification. Both the parametric and NPL posteriors target the same parameter, denoted as $\boldsymbol{\theta}^*$, which minimizes the KL divergence between the true distribution $F^*$ and the model distribution $F_\theta$ (Walker, 2013).

However, it is crucial to note that NPL utilizes a nonparametric prior, leading to posterior distributions on $\boldsymbol{\theta}^*$ with superior asymptotic properties compared to the regular Bayesian posterior when the model is misspecified. The key distinction lies in the fact that parametric Bayesian inference assumes the existence of $\boldsymbol{\theta}^*$ such that $F_{\boldsymbol{\theta}^*} = F^*$. On the other hand, NPL does not make this assumption, acknowledging model misspecification, and updates the posterior distribution $\pi_n(F)$ in a nonparametric manner. This results in more robust posterior inferences on $\boldsymbol{\theta}^*$ and asymptotically superior predictions, as observed practically (Fong et al., 2019).

To elaborate on the superior asymptotic properties, Holmes & Walker (2019) demonstrate that the Bayesian bootstrap posterior (having the same limit as NPL) asymptotically yields the sandwich covariance matrix, known for its robustness. Conversely, the parametric posterior does not achieve this asymptotically.

In terms of prediction, our focus is on the posterior predictive density $p_n(y) = \int f_{\boldsymbol{\theta}}(y)\pi_n(\boldsymbol{\theta})$, where $\pi_n$ represents either the Bayesian or NPL posterior. Theorem 1 of Lyddon et al. (2018) shows that asymptotically, the KL divergence between $F^*$ and $P_n$ is smaller for the NPL posterior compared to the Bayesian posterior. This asymptotic improvement is attributed to the robust sandwich covariance matrix (Müller, 2013).

