# OpenReview forum: "Enhancing Transfer Learning with Flexible Nonparametric Posterior Sampling"
_ICLR.cc/2024/Conference — ICLR 2024 poster_

### Official Review · Reviewer_ZDrq · 2023-10-27

**Soundness:** 3 good
**Presentation:** 3 good
**Contribution:** 3 good
**Rating:** 6
**Confidence:** 4

**Summary:**

In the context of transfer learning, the choice of the prior distribution for downstream data is crucial when employing Bayesian model averaging (BMA). Prior strategies have limitations in handling distribution shifts between upstream and downstream data. This paper introduces nonparametric transfer learning (NPTL) that addresses distribution shift issues within the framework of nonparametric learning. The nonparametric learning (NPL) method, which uses a nonparametric prior for posterior sampling, efficiently deals with model misspecification scenarios.

**Strengths:**

This paper proposes the use of nonparametric learning techniques to address the issue of distribution shift in transfer learning. The writing in the article is of high quality and is easy to understand. I partially reviewed the mathematical derivations in the paper, and the technical aspects appear to be sound.

**Weaknesses:**

I did not find any significant drawbacks in the paper, except for some confusion in the nonparametric learning section. Please see below for details.

**Questions:**

I feel confused about the authors' claim that "the NPL posterior is robust to model misspecification through the adoption of a nonparametric model and gives asymptotically superior predictions to the regular Bayesian posterior on $\theta$".

In my understanding, NPL is just another way to compute the posterior for $\theta$. In parametric Bayesian methods, a prior $p(\theta)$ is placed on $\theta$, and various methods are employed to compute the corresponding posterior $p(\theta|\mathcal{D})$. NPL, on the other hand, establishes a deterministic mapping between the data distribution $F$ and $\theta$ using MLE (as shown in Equation 2). Then, a prior $p(F|\alpha, F_\pi)$ is placed on the data distribution $F$ to analytically derive the posterior $p(F|\mathcal{D})$, which subsequently leads to the posterior $p(\theta|\mathcal{D}$).

Both of the above methods assume a parameterized model with parameters $\theta$. In other words, the assumed parameterized model is still susceptible to potential discrepancies with the true data distribution, which could result in model misspecification. Therefore, if my understanding is correct, NPL may not inherently address the issue of model misspecification.

---

> ### Author Response · Authors · 2023-11-11
> **Rebuttal by Authors**
>
> Thank you for your constructive and positive comment with an important question.
>
> __Q1 Is the NPL posterior robust to model misspecification compared to the regular Bayesian posterior on $\theta$?__
>
> Thank you to the reviewer for posing an important and insightful question. You are indeed correct that the parametric posterior and NPL posterior target the same parameter, that is $\theta^*$ which minimizes the KL divergence between the true $F^*$ and the model $F_\theta$ [Walker (2013)].
>
> However, we highlight that NPL elicits a nonparametric prior, which induces a
> posterior distributions on $\theta^*$ that has superior asymptotic properties to the regular Bayesian posterior when the model is misspecified. Intuitively, the reason for this subpar performance for parametric Bayes is that the posterior is computed assuming there exists $\theta^*$ such that $F_{\theta^*} = F^*$. On the other hand, NPL does not make this assumption (i.e. model misspecification is acknowledged), and updates the posterior distribution $\pi_n(F)$ in a nonparametric fashion. This in turn leads to more robust posterior inferences on $\theta^*$, as well as asymptotically superior predictions. This improvement in prediction is indeed observed practically as well [Fong et al. (2019)].
>
> We now outline what we mean specifically by superior asymptotic posteriors and predictions.  Lyddon et al. (2018) show that the Bayesian bootstrap posterior (which has the same limit as NPL) asymptotically has the sandwich covariance matrix, which is known to be robust. On the other hand, the parametric posterior does not obtain this variance asymptotically.
>
> For prediction, we are interested in the posterior predictive density,
> $p_n(y) = \int f_\theta(y) \pi_n(\theta)$, where $\pi_n$ is either the Bayesian or the NPL posterior. Theorem 1 of Lyddon et al. (2018) shows that asymptotically, the KL divergence between $F^*$ and $P_n$ is smaller for the NPL posterior compared to the Bayesian posterior. This asymptotic improvement is indeed due to the robust sandwich covariance matrix [Muller (2013)].
>
> Thank you again for highlighting this important point. We will add a detailed discussion of how NPL addresses model misspecification in the revision paper. And if there are no additional questions or uncertainties, we kindly request you to contemplate revisiting your evaluation to ensure it accurately reflects the situation.
>
> __References__
>
> [1] Walker, S. G. (2013). Bayesian inference with misspecified models. Journal of statistical planning and inference, 143(10), 1621-1633.
>
> [2] Lyddon, S. P., Holmes, C. C., & Walker, S. G. (2019). General Bayesian updating and the loss-likelihood bootstrap. Biometrika, 106(2), 465-478.
>
>
> [3] Lyddon, S., Walker, S., & Holmes, C. C. (2018). Nonparametric learning from Bayesian models with randomized objective functions. Advances in neural information processing systems, 31.
>
> [4] Müller, U. K. (2013). Risk of Bayesian inference in misspecified models, and the sandwich covariance matrix. Econometrica, 81(5), 1805-1849.
>
> [5] Fong, E., Lyddon, S., & Holmes, C. (2019, May). Scalable nonparametric sampling from multimodal posteriors with the posterior bootstrap. In International Conference on Machine Learning (pp. 1952-1962). PMLR.

---

> ### Author Response · Authors · 2023-11-20
> **Rebuttal response request**
>
> Your contribution to reviewing our paper is much appreciated. With the discussion period deadline nearing, could you kindly respond to our rebuttals? It would greatly assist us if you could indicate any further inquiries or uncertainties you might have regarding our paper.

---

> > ### Comment · Reviewer_ZDrq · 2023-11-20
> > **Response to authors' rebuttal**
> >
> > Thanks for the authors' feedback. I do not have any further concern and I am satisfied with authors' answers in this rebuttal. I raise the score to 6.

---

> > > ### Author Response · Authors · 2023-11-20
> > > **Official Comment by Authors**
> > >
> > > We're grateful for your encouraging and insightful feedback. In the revised paper, we will add a detailed discussion of how NPL addresses model misspecification.

---

### Official Review · Reviewer_izux · 2023-10-31

**Soundness:** 3 good
**Presentation:** 3 good
**Contribution:** 3 good
**Rating:** 6
**Confidence:** 3

**Summary:**

In these transfer learning scenarios, the prior distribution for downstream data becomes crucial in Bayesian model averaging (BMA). While previous works proposed the prior over the neural network parameters centered around the pre-trained solution, such strategies have limitations when dealing with distribution shifts between upstream and downstream data. This paper introduces nonparametric transfer learning (NPTL), a flexible posterior sampling method to address the distribution shift issue within the context of nonparametric learning. Through extensive empirical validations, the authors demonstrate that the approach surpasses other baselines in BMA performance.

**Strengths:**

(1) The authors maintain high quality of presentation. The motivation and algorithms are clearly explained.

(2) Extensive experiments and ablation studies are conducted in this paper. The authors considered different model architectures, datasets, evaluation metrics, tasks.

**Weaknesses:**

(1) A detailed discussion of limitations is lacking. A possible aspect could be the computation cost. A careful discussion of pros and cons could be very helpful to the community.

**Questions:**

(1) Following the experiments of [1], could the proposed method also uses self-supervised pretrained models and include experiments about segmentation?

references:

[1] Shwartz-Ziv, Ravid, Micah Goldblum, Hossein Souri, Sanyam Kapoor, Chen Zhu, Yann LeCun, and Andrew G. Wilson. "Pre-train your loss: Easy bayesian transfer learning with informative priors." Advances in Neural Information Processing Systems 35 (2022): 27706-27715.

---

> ### Author Response · Authors · 2023-11-14
> **Rebuttal by Authors**
>
> Thank you for your constructive and positive comments. We will summarize and respond to the questions below:
>
> __W1 Discussion of limitations is lacking.__
>
> As you pointed out, the apparent downside of the suggested NPTL approach is the extra training cost. For instance, applying the proposed NPTL algorithm involves additional computational expenses to obtain $f_{\text{probed}}$ through linear probing for the provided downstream data. Nevertheless, the computational cost associated with linear probing is minimal in contrast to training the complete network. We will add a discussion on such limitations regarding computational costs in the next version of the paper.
>
> __Q1 (1) Using self-supervised pre-trained models__
>
> Thank you for posing an important question about constructing pre-trained models. Since the informative base measure $F_\pi(x, y)$ could be any probability measure representing our initial assumption about $F_0$ based on our prior knowledge of the actual data distribution, we have the flexibility to employ linear probed self-supervised pre-trained models as our model $f_{\text{probed}}. Indeed, our preliminary experimental results on Caltech101, utilizing ResNet-50-SimCLR as the pre-trained model, are as follows: Ensemble + L2SP attains ACC=0.875 and NLL=0.441, whereas NPTL attains ACC=0.879 and NLL=0.410. We believe that integrating the outcomes of self-supervised pre-training will undoubtedly enhance the paper, and we are grateful for constructive feedback. Additional results regarding self-supervised pre-training will be available in the camera-ready version.
>
> __Q1 (2) Experiments on semantic segmentation__
>
> Understanding semantic segmentation as pixel-level classification would simplify the implementation of the proposed NPTL in semantic segmentation experiments. More precisely, we first obtain $f_{\text{probed}}$ by exclusively training additional modules for semantic segmentation tasks while keeping the fixed (self-supervised) pre-trained backbone network. Subsequently, we proceed to train the entire semantic segmentation model, incorporating the proposed NPTL term.
>
> In this manner, there is no obstacle to the conceptual application of our NPTL methodology to semantic segmentation tasks. We are of the opinion that providing additional experimental results for semantic segmentation, as per your suggestion, will enhance the emphasis on our contribution. Nevertheless, due to practical limitations like restricted computational resources and time constraints, we regretfully acknowledge that we may be unable to provide additional experiments on semantic segmentation during the rebuttal period. Even if the results are not ready by then, we will be committed to incorporating them into the camera-ready version. We earnestly request your understanding and consideration of this matter for your assessment.

---

> ### Author Response · Authors · 2023-11-21
> **Additional Results for semantic segmentation experiments**
>
> Following your constructive suggestion, we expanded our experiments on semantic segmentation using the PASCAL VOC 2012 dataset, building upon [1]. Our backbone model was the pre-trained ResNet-50, while we adopted the DeepLabv3+ model as our complete model. Our experiments, apart from our model, were consistent with those in [1]. Table R.3 presents conclusive evidence that NPTL surpasses other baseline models.
>
> __Table R.3.__ Additional BMA Mean IoU performance of (a) SGLD with Non-Learned Prior, (b) SGLD with Learned Prior Supervised, (c) SGLD with Learned Prior SSL, and (d) NPTL with ResNet-50.
> |  Dataset   | Backbone | (a) | (b) | (c) |(d)|
> | :-: | :-:          | :-:          | :-:          | :-:          |:-:          |
> | PASCAL VOC 2012 | ResNet-50 | 73.16 | 73.72 | 74.15 | 74.72|
>
> It would greatly assist us if you could indicate any further inquiries or uncertainties you might have regarding our paper.
>
> __References__
>
> [1] Shwartz-Ziv, Ravid, Micah Goldblum, Hossein Souri, Sanyam Kapoor, Chen Zhu, Yann LeCun, and Andrew G. Wilson. "Pre-train your loss: Easy bayesian transfer learning with informative priors." Advances in Neural Information Processing Systems 35 (2022): 27706-27715.

---

> ### Comment · Reviewer_izux · 2023-11-22
> **Thank you for the response**
>
> I have read the response and I will keep my score.

---

> ### Author Response · Authors · 2023-11-22
> **Official Comment by Authors**
>
> Thank you for your positive and constructive comments. We'll add our additional experiment results in new version of our paper.

---

### Official Review · Reviewer_bi4C · 2023-10-31

**Soundness:** 3 good
**Presentation:** 3 good
**Contribution:** 3 good
**Rating:** 6
**Confidence:** 4

**Summary:**

The paper studies the applicability of nonparametric transfer learning (NPTL) in the context of Bayesian NNs and transfer learning. The general idea is that having pre-trained models based on NNs, once can obtain good performance on Bayesian model averaging (BMA) prediction on different tasks. For instance, pre-training on Imagenet and testing performance on CIFAR or similar vision datasets.

**Strengths:**

In general, I think it is a good paper with positive ideas and contributions. I particularly see the interest behind the application of NPTL in this context for transfer learning. To me, the paper is clear in the details concerning the sampling methodology, and perhaps not that much in the problems related to scalability or computational cost (see my comments below).

**Weaknesses:**

In my opinion, I think there are several points that are not clear enough while reading the manuscript and they could be also potential weaknesses of the method.

[w1] --- Access to the posterior distribution given pre-trained models. In general, I see the idea, but it is not entirely clear to me how we can get posterior samples from any model that has been pre-trained without a particular prior before. Are we in the MAP solution? similarly to the Laplace approximation. Are there conditions or requirements for pre-training the models?

[w2] --- I appreciate the details and the sincere comments on the heuristics used and so on. However, I do not get a good feeling on the scalability and the computational cost. Is really the methods playing a role given huge models with large number of parameters? or is it just bc the pre-trained models are doing still well on similar vision tasks. In that regard, I'm not entirely convinced by the empirical results.

**Questions:**

see my prev. comments

---

> ### Author Response · Authors · 2023-11-10
> **Review confirmation request**
>
> We appreciate your dedication to reviewing our paper. However, it seems there might be a confusion with other paper, as we don't have any anonymous references in our paper or reliance on a theory submitted to ICLR 2024 in another submission. Could you please double-check and confirm your observations? Thank you.

---

> ### Author Response · Authors · 2023-11-15
> **Rebuttal by Authors**
>
> As, we have identified what seems to be the accurate review for our paper in this link: https://openreview.net/forum?id=AweVGJeW47&noteId=4ovgy5lYGf.
>
> We write responses based on that review.
>
> Thank you for your constructive and positive comment. We will summarize and respond to the questions below:
>
> __W1 How we can get posterior samples from any model that has been pre-trained without a particular prior before. Are there conditions or requirements for pre-training the models?__
>
> Thank you for pointing out an important question. As we commented on the question from reviewer izux, there is no restriction to constructing pre-trained models. Since the informative base measure $F_\pi(x, y)$ could be __any probability measure representing our initial assumption about $F_0$ based on our prior knowledge of the actual data distribution__, we have the flexibility to employ any type of pre-trained models that contain information of generating process of the upstream dataset (MAP solution from supervised tasks or solution from self-supervised).
>
> __W2 (1) Is the method really playing a role given huge models with a large number of parameters?__
>
> As you pointed out, there is a trend in modern transfer learning to assume large pre-trained models on extensive upstream data. Consequently, it would be essential to assess the scalability of transfer learning approaches. To this end, we conducted experiments employing the sizable ViT-B/16 and RoBERTa-Base models (Specifically, ResNet-50 has approximately 23 million parameters, whereas ViT-B/16 and RoBERTa-Base have 86 million and 123 million parameters, respectively). We believe that our experiments using ViT and RoBERTa adequately validate the scalability of the proposed NPTL methodology.
>
> __W2 (2) Is it just because the pre-trained models are doing still well on similar vision tasks?__
>
> Your question brings up an intriguing point. Indeed, a well-pretrained model tends to perform effectively across related tasks. Yet, if the prior distribution isn't suitable, it can significantly impact the performance of the Bayesian model averaging (BMA) with posterior samples. To demonstrate this, we showcase the BMA performance using the zero-mean isotropic Gaussian prior, denoted as $\mathcal{N}(0,\sigma^2 * I)$. The empirical results in Table R.1. and Table R.2. distinctly illustrate how an unsuitable prior distribution can markedly degrade performance.
>
> __Table R.1.__ Additional BMA accuracy performance of (a) SGHMC with zero-mean isotropic Gaussian prior and (b) NPTL with ResNet-20x4.
> |     | CIFAR-10 | CIFAR-100 | Food-101 | Dogs-120 |
> | :-  | :-:          | :-:          | :-:          | :-:          |
> | (a) | 0.872 ± 0.000 | 0.625 ± 0.001 | 0.550 ± 0.001 | 0.235 ± 0.001 |
> | (b) | 0.964 ± 0.001 | 0.818 ± 0.000 | 0.644 ± 0.002 | 0.634 ± 0.001 |
>
> __Table R.2.__ Additional BMA negative log-likelihood performance of (a) SGHMC with zero-mean isotropic Gaussian and (b) NPTL with ResNet-20x4.
> |     | CIFAR-10 | CIFAR-100 | Food-101 | Dogs-120 |
> | :-  | :-:          | :-:          | :-:          | :-:          |
> | (a) | 0.381 ± 0.001 | 1.321 ± 0.001 | 1.733 ± 0.003 | 3.071 ± 0.002 |
> | (b) | 0.102 ± 0.001 | 0.606 ± 0.002 | 1.347 ± 0.001 | 1.297 ± 0.003 |

---

> > ### Comment · Reviewer_bi4C · 2023-11-17
> > **Apologies and correction of "swapped" review**
> >
> > I sincerely want to apologize to the authors (AC, PCs and the rest of reviewers) for my mistake on the initial review, which I accidentally swapped with another submission with "similar" ID number. Since my original (draft) review (out of the openreview system) was rather positive (soundness, presentation, and contribution were good in my opinion with an acceptance score recommendation), I also want to apologize to the authors for receiving a review with a reject recommendation, particularly, if this has caused a bad impression of the reviewing process of ICLR, which I remark is still rigorous and aware of the mistake I accidentally committed. Last but not least, I want to remark authors' effort to find the correct review in the openreview blind system and make a rebuttal according to the comments I included there.
> >
> > ---------------------------------------
> >
> > Just for future reference, the correct review of this paper was initially this one and is the one the authors used for the rebuttal (it is also updated as main review)
> >
> > **Summary:** The paper studies the applicability of nonparametric transfer learning (NPTL) in the context of Bayesian NNs and transfer learning. The general idea is that having pre-trained models based on NNs, once can obtain good performance on Bayesian model averaging (BMA) prediction on different tasks. For instance, pre-training on Imagenet and testing performance on CIFAR or similar vision datasets.
> >
> > **Strengths:** In general, I think it is a good paper with positive ideas and contributions. I particularly see the interest behind the application of NPTL in this context for transfer learning. To me, the paper is clear in the details concerning the sampling methodology, and perhaps not that much in the problems related to scalability or computational cost (see my comments below).
> >
> > **Weaknesses:** In my opinion, I think there are several points that are not clear enough while reading the manuscript and they could be also potential weaknesses of the method.
> >
> > [w1] --- Access to the posterior distribution given pre-trained models. In general, I see the idea, but it is not entirely clear to me how we can get posterior samples from any model that has been pre-trained without a particular prior before. Are we in the MAP solution? similarly to the Laplace approximation. Are there conditions or requirements for pre-training the models?
> >
> > [w2] --- I appreciate the details and the sincere comments on the heuristics used and so on. However, I do not get a good feeling on the scalability and the computational cost. Is really the methods playing a role given huge models with large number of parameters? or is it just bc the pre-trained models are doing still well on similar vision tasks. In that regard, I'm not entirely convinced by the empirical results.

---

> ### Comment · Reviewer_bi4C · 2023-11-17
> **Response to authors' rebuttal.**
>
> Thanks to the authors for the answers to my questions included in the (correct) review. Some short comments to their points to make a follow up message on this rebuttal period.
>
> [W1] --- Ok, I see it now, also after section (2) and using section 2.3 and 2.4 from Fong et al. (2019). It makes sense to me, thank you.
>
> [W2A] --- I agree that the order of parameters considered makes sense given the proposed method and approach for probabilistic transfer learning. To me sounds convincing. Additionally, the sort of models considered is also in line with other recent papers doing transfer learning approaches in similar scenarios like in (Matena and Raffel, NeurIPS 2022). Thanks for clarifying this out.
>
> [W2B] --- Thanks for running out additional experiments to address my question. I see the point of author's response and makes sense to me. I also liked the experiment conducted. It would be definitely interesting to test this sort of impact with pre-trained models in the future. But I also understand it could be out of the paper's current scope.
>
> In general, I am satisfied with authors' contributions and answers in this rebuttal. Therefore, I will raise my score accordingly.
>
>
>
> Matena and Raffel. Merging Models with Fisher-Weighted Averaging. NeurIPS 2022

---

> > ### Author Response · Authors · 2023-11-18
> > **Official Comment by Authors**
> >
> > Thanks once more for your valuable feedback and insightful discussions. We'll integrate these discussions into the updated version of our paper.

---

### Official Review · Reviewer_vw7S · 2023-11-01

**Soundness:** 3 good
**Presentation:** 3 good
**Contribution:** 3 good
**Rating:** 6
**Confidence:** 3

**Summary:**

This paper introduces nonparametric transfer learning (NPTL), which uses a Dirichlet process prior with centering measure $F_{\pi}$ defined by downstream samples and their outputs from a linear-probed model. Then each posterior sample defines a weighted loss of the downstream samples and pseudo samples from the linear-probed model. The weighted loss is optimized for each posterior sample. Then Bayesian model averaging is performed over $M$ posterior samples. Extensive experiments show the superior performance of the proposed method over traditional Bayesian sampling methods.

**Strengths:**

Using downstream samples and a linear prob model to construct prior for transfer learning is an interesting idea, and with Dirichlet Processes, the posterior gives a simple form. The paper provides extensive numerical experiments to support the superior performance of the proposed method. The limitation of Bayesian model averaging: the computational cost is also properly discussed.

**Weaknesses:**

I found some details of the proposed algorithm a bit confusing, especially in the context of transfer learning. I want to clarify the following points

1. For the linear prob model, my understanding is that $\phi^*$ is from the pre-trained model, how do we get $W^*$, is it obtained by only fitting the last fully connected layer on the downstream task?

2. In step 7 of algorithm 1, how is $\theta$ initialized, randomly or by $(\phi^*, W^*)$, i.e. is the pre-trained model only used to create pseudo samples for the prior, or is it also used to initialize parameter as well. Looking at the training setting, e.g. ResNet-50, with a cosine learning rate decay schedule, it looks like the parameters are trained from scratch, otherwise, I imagine the large initial learning rate would drive parameters away from the initialization.

3. For the SGHMC baseline, how is the prior specified and how does the pre-trained model help the SGHMC method? And how does the pre-trained model help the ensemble baseline?

**Questions:**

Please see the questions in the weakness above. Some minor questions

1. In section 2.1 paragraph 1. the zero mean isotropic Gaussian is called a non-informative prior. If I remember correctly, the non-informative prior usually refers to something else.

2. For the Bayesian inference, one criterion of setting prior can be ensuring posterior concentration, e.g. in the sense of theorem 2.1 in [1]. Can the authors comment a bit on how the proposed prior related to those concepts?

### Reference
[1] Ghosal, Subhashis, Jayanta K. Ghosh, and Aad W. Van Der Vaart. "Convergence rates of posterior distributions." Annals of Statistics (2000): 500-531.

---

> ### Author Response · Authors · 2023-11-13
> **Rebuttal by Authors**
>
> Thank you for your positive and constructive reviews and comments. We will summarize and respond to individual comments below:
>
> __W1 How do we get W*?__
>
> Your understanding is correct. We specifically fit the last fully-connected layer for the downstream task, ensuring that our $(\phi^\ast, W^\ast)$ retains the essence of the upstream data-generating process in $\phi^\ast$ and transfers the knowledge from the upstream through $W^\ast$.
>
> __W2 Initialization and learning rate.__
>
> Our proposed posterior sampling method is tailored for transfer learning scenarios, and we initiated the model parameters as $\mathbf{\theta} = (\phi^\ast, W)$, where $\phi^\ast$ represents a pre-trained feature extractor parameter, and $W$ is the randomly initialized task-specific head parameter. The linear-probed head parameter $W^\ast$ is solely utilized in constructing the informative base measure for NPTL (although it could potentially be initialized with $W^\ast$, we assert that using the same random initialization for all methods offers a more equitable basis for experimental comparison).
>
> As you pointed out, when fine-tuning from a pre-trained solution, it is crucial to be cautious about avoiding substantial deviations from it. In our ResNet-50 experiments, we opted for a learning rate of 0.01, smaller than the conventional value used in from-scratch training (e.g. 0.1 in practice). Consequently, we did not encounter situations where the model significantly deviated from the pre-trained initialization. Additionally, when testing larger learning rate values such as 0.1 and 0.03, we observed a significant performance decline in line with your concern, indicating a departure from the pre-trained initialization.
>
> __W3 For the SGHMC baseline, how is the prior specified and how does the pre-trained model help the SGHMC method? And how does the pre-trained model help the ensemble baseline?__
>
> For the SGHMC baseline, we employed two distinct weight prior distributions as outlined below:
> 1. **L2SP:** Gaussian prior with a mean set as the pre-trained weight and a variance of $\sigma^2 * I$, where $I$ represents the identity matrix.
> 2. **PTYL:** Gaussian prior with a mean derived from the pre-trained weight, further trained on the upstream dataset (acknowledged as challenging to utilize, as emphasized in the paper). The variance is a non-diagonal matrix $\Sigma$ constructed with empirical covariance using weights gathered during additional training on the upstream dataset.
>
> Considering the context of posterior sampling in the transfer learning scenario, both the ensemble baselines and the SGHMC method adopt the same initialization approach. Specifically, we initialize our method and all the baselines with pre-trained feature extractor parameters $\phi$ and randomly initialized task-specific parameters $W$.
>
>
> __Q1 The zero mean isotropic Gaussian is called a non-informative prior.__
>
> Thank you for bringing this confusion to our attention. We will seek more suitable language and make the necessary corrections.
>
> __Q2 For the Bayesian inference, one criterion of setting prior can be ensuring posterior concentration. Can the authors comment a bit on how the proposed prior related to those concepts?__
>
> Thank you for highlighting the importance of posterior concentration. As NPL is based on the Dirichlet Process, we can leverage standard results. In fact, more can be said than the posterior contraction rate - for example, Theorem 12.2 of Ghosal & Van der Vaart (2017) gives a Bernstein-von Mises theorem for the Dirichlet process.
>
> __References__
>
> [1] Ghosal, S., & Van der Vaart, A. (2017). Fundamentals of nonparametric Bayesian inference (Vol. 44). Cambridge University Press.

---

> > ### Comment · Reviewer_vw7S · 2023-11-20
> >
> > I appreciate the authors' detailed response. The clarification of the algorithm makes sense in the transfer learning context. I don't have further concerns, and I will keep my positive rating.

---

> ### Author Response · Authors · 2023-11-20
> **Rebuttal response request**
>
> Your contribution to reviewing our paper is much appreciated. With the discussion period deadline nearing, could you kindly respond to our rebuttals? It would greatly assist us if you could indicate any further inquiries or uncertainties you might have regarding our paper.

---

> ### Author Response · Authors · 2023-11-20
> **Official Comments by Authors**
>
> Thank you for your valuable and positive comments. We will make the correction in the revised version of our paper.

---

### Author Response · Authors · 2023-11-13
**General Response by Authors**

We thank all reviewers for their valuable and constructive comments. We are pleased that they found the paper clearly written with good organization (R-izux, R-ZDrq, R-bi4C) and well-motivated (R-izux, R-ZDrq, R-bi4C). They also found the proposed method novel (R-vw7S, R-bi4C) and that the paper presents extensive experiments and ablation studies (R-vw7S, R-izux, R-bi4C). Moreover, they have emphasized the soundness of the mathematical derivations in the paper (R-ZDrq).

---

### Meta-Review · Area_Chair_UscH · 2023-12-10

**Metareview:**

All reviewers found this work solid and of broad interest. The claims are supported and the rebuttal clarified the reviewers' concerns. The additional experimental results were appreciated. I would encourage the authors to incorporate the clarifications of the algorithm and other discussion points they provided during the rebuttal. I would also encourage the authors to integrate the additional experimental results in the main paper or an appendix as these are worthwhile results to report.

As a side note, I would like to apologise for the confusing review initially posted by one of the reviewers. Luckily, the error has been corrected in time and did not impact the evaluation of this work.

**Justification For Why Not Higher Score:**

All reviewers were in agreement about the evaluation of this paper (score of 6). This is solid work and valuable to the community, but there are no remarkable results or contributions that would call for a spotlight (or oral).

**Justification For Why Not Lower Score:**

All reviewers agreed this was a solid paper that should be accepted.

---

### Decision · Program_Chairs · 2024-01-16

Accept (poster)